# Scars from protected zero modes and beyond in $U(1)$ quantum link and quantum dimer models

**Saptarshi Biswas[1,2], Debasish Banerjee[3,4] and Arnab Sen[5⋆]**

1 Department of Physical Sciences, Indian Institute of Science Education and
Research-Kolkata, Mohanpur, Nadia 741246, India
2 Department of Physics & Astronomy, Northwestern University, Evanston, IL 60208, USA
3 Theory Division, Saha Institute of Nuclear Physics, 1/AF Bidhan Nagar,
Kolkata 700064, India
4 Homi Bhabha National Institute, Training School Complex, Anushaktinagar,
Mumbai 400094, India
5 School of Physical Sciences, Indian Association for the Cultivation of Science,
Kolkata 700032, India

⋆ tpars@iacs.res.in

## Abstract

We demonstrate the presence of anomalous high-energy eigenstates, or many-body scars, in $U(1)$ quantum link and quantum dimer models on square and rectangular lattices. In particular, we consider the paradigmatic Rokhsar-Kivelson Hamiltonian $H = \mathcal{O}_{\mathrm{kin}} + \lambda \mathcal{O}_{\mathrm{pot}}$ where $\mathcal{O}_{\mathrm{pot}}$ ($\mathcal{O}_{\mathrm{kin}}$) is defined as a sum of terms on elementary plaquettes that are diagonal (off-diagonal) in the computational basis. Both these interacting models possess an exponentially large number of mid-spectrum zero modes in system size at $\lambda = 0$ that are protected by an index theorem preventing any mixing with the nonzero modes at this coupling. We classify different types of scars for $|\lambda| \lesssim \mathcal{O}(1)$ both at zero and finite winding number sectors complementing and significantly generalizing our previous work [Banerjee and Sen, Phys. Rev. Lett. **126**, 220601 (2021)]. The scars at finite $\lambda$ show a rich variety with those that are composed solely from the zero modes of $\mathcal{O}_{\mathrm{kin}}$, those that contain an admixture of both the zero and the nonzero modes of $\mathcal{O}_{\mathrm{kin}}$, and finally those composed solely from the nonzero modes of $\mathcal{O}_{\mathrm{kin}}$. These scars have tell-tale energies such as (non-zero) integers and irrationals like $\pm\sqrt{2}$ at $\lambda = 0$ or $n_1\lambda \pm n_2$ at $\lambda \neq 0$ where both $n_1, n_2$ are integers. We give analytic expressions for certain "lego scars" for the quantum dimer model on rectangular lattices where one of the linear dimensions can be made arbitrarily large, with the building blocks (legos) being composed of emergent singlets and other more complicated entangled structures.



# 1   Introduction

Generic non-integrable many-body quantum systems are expected to locally equilibrate to a thermal ensemble, whose temperature is set by the energy density of the initial state, when evolved under the unitary time dynamics generated by their own Hamiltonian [1]. This expectation is based on the eigenstate thermalization hypothesis (ETH) which states that high-energy eigenstates of such interacting systems appear locally thermal [2–5]. Many-body localization is a well-known mechanism [6] to evade this paradigm by introducing strong disorder which leads to an emergent integrability under certain conditions [7,8]. An open question in the field is to understand whether self-thermalization can be evaded in translationally invariant non-integrable systems without any disorder.

Recently, a class of anomalous high-energy eigenstates have been identified in a variety of quantum systems ranging from spins [9–28], bosons [29–31], fermions [32–39], lattice gauge theories [40–42] as well as in driven quantum matter [43–50] which distinguish themselves from the ETH band by possessing very low entanglement entropy. This constitutes a weak breaking of ergodicity whereby initial states that have significant overlap with such anomalous high-energy eigenstates dubbed quantum many-body scars, evade thermalization. This is distinct from many-body localized systems where ergodicity is broken strongly. While not all the mechanisms of such scarring phenomena are well understood, their occurrence in a wide class of models might point to the possibility of a more general underlying principle to evade the ETH in disorder-free interacting theories.

Some of the questions that motivate this particular work are as follows:

- Models with a constrained Hilbert space have played a central role in the study of quantum scars, starting with the archetypal PXP model [51,52] which is realized experimen-

tally using Rydberg atoms [53]. Constrained Hilbert spaces also arise in Hamiltonian formulations of lattice gauge theories (LGTs) [54] since gauge-invariant states necessarily satisfy a Gauss law. In fact, the PXP model maps exactly to a lattice Schwinger model where the gauge fields are coupled with staggered fermions in one dimension [41]. It is then natural to ask whether quantum scars appear in gauge theories without dynamical matter fields and can the underlying scarring mechanism be fundamentally different from that of the PXP model? Specifically, scars in the PXP model are approximately equidistant in energy and their number scales extensively with system size [9,10]. Both features arise from an emergent $SU(2)$ spin description of a small subspace of the Hilbert space. Identification of completely different mechanism(s) will benefit from analogous information about the pure gauge theory scars.

- Several interacting models satisfy an index theorem [55] (see also, Ref. [9]) due to the intertwining of a chiral and a lattice inversion symmetry that leads to the presence of exact zero modes whose number scales exponentially in system size in a many-body spectrum that is symmetric around zero energy. While these mid-spectrum zero modes are expected to satisfy the ETH, recent works [42, 56] have highlighted the intriguing possibility that special linear combinations of these zero modes exist as anomalous states. However, dynamical signatures of such anomalous zero modes in simple initial states may be obscured by the presence of the other exponentially many non-anomalous zero modes in the spectrum. This is because the usual diagonal ensemble $\rho_{\mathrm{DE}} = \sum_E |\langle E|\mathrm{in}\rangle|^2 |E\rangle\langle E|$ (where $|\mathrm{in}\rangle$ and $|E\rangle$ denote an initial state and an energy eigenstate respectively) that captures the time-average of any local observable starting from an initial state is no longer applicable when the spectrum has exact degeneracies. Instead, one must first diagonalize a particular observable in the basis of the degenerate eigenstates and then use the corresponding diagonal ensemble.

  Can further non-commuting interaction terms be added to such interacting models with large nullspaces to break the index theorem but stabilize these (or a subset thereof) anomalous zero modes as true eigenstates while the rest of the zero modes hybridize with the nonzero modes? Ref. [42] showed this to be the case for a particular $U(1)$ LGT, the $S = 1/2$ quantum link model [57] on a ladder geometry by starting with a strongly interacting limit of the model where an exponentially large number of protected zero modes emerge due to the index theorem and then adding a gauge-invariant non-commuting term in the Hamiltonian to break the index theorem. However, several open questions remain from that initial study. Are there other models where this mechanism can be shown to be at work? Can the existence of these anomalous zero modes be shown analytically in the thermodynamic limit for some non-integrable model(s)? Are there other types of scars generated in such models apart from these anomalous zero modes and if so, then how to classify and understand them?

In this paper, we attempt to address the above questions by exploring the rich variety of scars in the $U(1)$ quantum link and quantum dimer models with a Rokhsar-Kivelson Hamiltonian on finite rectangular geometries with periodic boundary conditions in both directions. Such models have a venerable history in the studies of strongly correlated systems and are also examples of compact Abelian LGTs without any matter fields. The quantum dimer model is widely used in the context of frustrated antiferromagnets and high-temperature superconductors [58,59]. The quantum link model, on the other hand, is well known as a microscopic model which realizes the low-energy gauge-invariant subspace of quantum spin-ice [60,61], while in the context of particle physics it has been shown to realize novel crystalline confining phases with fractionalization of electric flux [62].

We would like to emphasize that the mechanism of scarring observed for the anomalous

zero modes in these models is consistent with the "order-by-disorder" paradigm [1], but realized in the Hilbert space as initially proposed in Ref. [42]. The exponential zero mode degeneracy observed for the operator $\mathcal{O}_{\text{kin}}$ gets lifted due to the addition of another non-commuting operator $\mathcal{O}_{\text{pot}}$ (both operators are introduced in Sec. 2). However, certain special combinations of these zero modes that are *simultaneous* eigenstates of both the operators are also created in the process. These eigenstates turn out to be much more localized in the Hilbert space compared to the other zero modes of $\mathcal{O}_{\text{kin}}$ and have anomalous physical properties. This mechanism is fundamentally different from that of Ref. [65] where quantum scars were demonstrated previously for a class of quantum dimer models on the kagome lattice based on a method to embed specific target states in the middle of a many-body spectrum [66].

The rest of the paper is arranged as follows. We start with a self-contained introduction to these models and their relation to Wilson's lattice gauge theory in Sec. 2, and discuss the global symmetries in Sec. 3, along with the index theorem that is valid at one particular coupling, $\lambda = 0$, where both models remain strongly interacting. To explore the properties of scars, we use large-scale symmetry-decomposed exact diagonalizations explained in Sec. 4. We use histograms of density of states to explicitly show the anomalously large number of the zero modes in different topological sectors when the index theorem holds. Using the level statistics obtained from the eigenspectra of the two models, their ergodic nature is also demonstrated for interesting parameter regimes. In Sec. 5, we identify different types of scars by using their entanglement properties, as well as via local correlation functions and show that the anomalous eigenstates occurring in the $U(1)$ quantum link model and the quantum dimer model can be classified into various different classes. These classes are most naturally viewed by expressing the scars in terms of the zero and the nonzero modes of the operator $\mathcal{O}_{\text{kin}}$ (which is introduced in Sec. 2). A numerical algorithm to isolate scars directly from the spectrum is explained in Sec. 6, together with a tabulation of the different types of scars and their degeneracies. We study the properties of each of the classes of the scars for both the quantum link model and the quantum dimer model in Sec. 7, extending far beyond the initial results of Ref. [42]. We find that all these scars have very characteristic energies that are either non-zero integers and simple irrationals like $\pm\sqrt{2}$ at the coupling $\lambda = 0$ or $n_1\lambda \pm n_2$, with $n_1$ and $n_2$ both being integers, at any $\lambda \neq 0$. This energy structure immediately highlights that the scarring mechanism here must be different from "PXP-type scars" where towers of exceptional states with nearly equidistant eigenenergies were identified. While some of these scars are eigenstates only at the specific coupling, $\lambda = 0$, where the aforementioned index theorem holds, there are other scars which remain eigenstates at any $\lambda$ or specifically at $\lambda \neq 0$. In Sec. 8 we explore the remarkable result, that a class of anomalous zero modes, which we call as "lego scars", can be proven to be exact eigenstates of the quantum dimer model at any coupling for certain rectangular geometries where one of the linear dimensions is kept fixed while the other can be made arbitrarily large with the number of these lego scars diverging exponentially in system size.

## 2 Quantum Link and Dimer Models

We first introduce the microscopic Hamiltonians and the constrained Hilbert space of the quantum link and the quantum dimer models, as well as explain how they are related to each other and to Wilson's lattice gauge theories. The physical motivation for such gauge theories is also emphasized.

As is well known, the Wilson version of an Abelian lattice gauge theory considers quantum

---

[1]"Order-by-disorder" mechanism is well-known in the context of frustrated magnetism [63,64] where thermal or quantum fluctuations lift the exponentially large degeneracy of classical ground states and induces order.

rotors placed on the links xy joining two neighboring sites x and y [67] (see Fig. 1). The gauge field operator is denoted as $U_{xy}$ and the corresponding electric flux as $E_{xy}$. In the electric flux basis (labelled by integer fluxes $0, \pm 1, \pm 2, \cdots$), the $U_{xy} = L_{xy}^+$ and $U_{xy}^\dagger = L_{xy}^-$ are the raising and lowering operators of electric flux which satisfy the canonical commutation relations:

$$[E_{xy}, U_{vw}] = U_{xy}\delta_{xv}\delta_{yw}\,; \qquad [E_{xy}, U_{vw}^\dagger] = -U_{xy}^\dagger\delta_{xv}\delta_{yw}\,. \tag{1}$$

The magnetic part of the Hamiltonian is composed of $n$-body interactions among the quantum rotors, where $n$ is the number of links around a closed loop of the lattice. For the square lattice, the elementary plaquette operator is $U_\square = U_{xy}U_{yz}U_{zw}^\dagger U_{wx}^\dagger$, where x, y, z, w are the four sites at the corner of a plaquette, see Fig. 1. In addition, the electric flux energy also contributes to the Hamiltonian:

$$H_{\text{Wilson}} = \mathcal{O}_{\text{kin}} + \frac{g^2}{2}E_{\text{flux}} = -\sum_\square \left(U_\square + U_\square^\dagger\right) + \frac{g^2}{2}\sum_{xy}E_{xy}^2\,. \tag{2}$$

The local Gauss law operator is $G_r = (\nabla \cdot E)_r = E_{ra} + E_{rb} - E_{rc} - E_{rd}$, where a, b, c, d are the forward-x, forward-y, backward-x, and backward-y neighbors respectively (see Fig. 1). Since the Gauss law commutes with the Hamiltonian $[H, G_r] = 0$, it breaks the Hilbert space into (exponentially) many superselection sectors. The local $U(1)$ invariance is due to the fact that one can use an angle $\theta_r = (0, 2\pi]$ locally to define the unitary transformation $V = \prod_r \exp(iG_r\theta_r)$ under which the Hamiltonian remains invariant: $H_{\text{Wilson}} = VH_{\text{Wilson}}V^\dagger$.

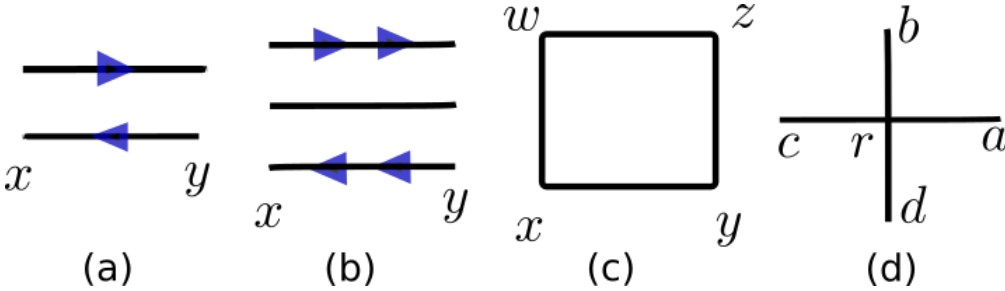

Figure 1: The gauge fields on links xy are the basic degrees of freedom of the models under consideration. From left to right: (a) two electric flux states of spin $S = \frac{1}{2}$ and (b) three states of spin $S = 1$ quantum link model, (c) the plaquette xyzw on a square lattice, and (d) Gauss' law on the square lattice relates the electric flux of the four links ra, rb, rc, and rd touching the site r. The Wilson limit is realized for $S \to \infty$.

Quantum rotors have an infinite dimensional Hilbert space at each link. A conceptual novelty is to regulate the local Hilbert space using quantum spin operators instead, such that there are only $(2S + 1)$ states for the gauge fields at a local site. This so-called quantum link formulation has been explored for both in high energy [57, 68, 69] and in condensed matter theory [60, 61, 70] communities. More recently quantum link models have been realized in quantum simulators [41, 53, 71, 72] and computers [73, 74] to explore the physics of gauge theories [75]. With quantum spins-$S$, the electric flux is represented by the $S^z$ (see Fig. 1), while the gauge fields are the raising and lowering operators of the flux: $E_{xy} = S^z$; $U_{xy} = S_{xy}^+$; $U_{xy}^\dagger = S_{xy}^-$. Since one is dealing with a finite dimensional Hilbert space, the gauge field operators are no longer unitary but satisfy $[U_{xy}, U_{vw}^\dagger] = 2E_{xy}\delta_{xv}\delta_{yw}$. The commutation relations between the gauge fields and the electric fluxes are unchanged, and thus $[G_r, U_\square] = 0$. The local $U(1)$ symmetry thus works as in the Wilson version, while the different Hilbert space gives rise to novel physics beyond the Wilson theory. It is possible to recover the Wilson theory in the limit of large spin representation-$S$ [76, 77].

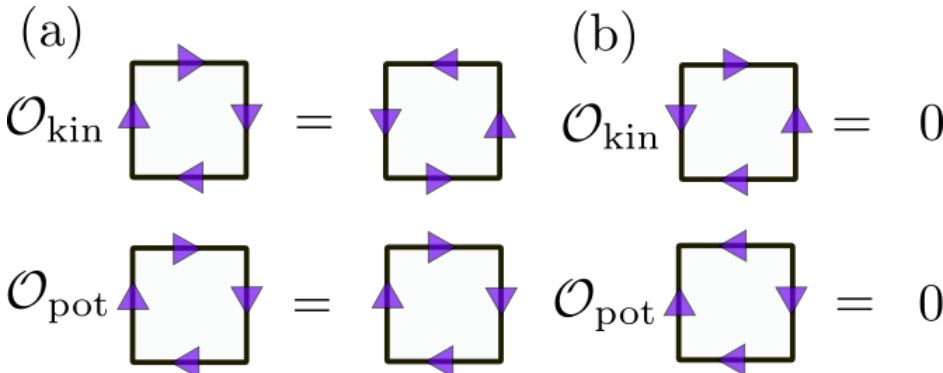

Figure 2: Pictorial representation of $\mathcal{H}_{\text{QLM}}$ and $\mathcal{H}_{\text{QDM}}$, both of which are composed of the operators $\mathcal{O}_{\text{kin}}$ and $\mathcal{O}_{\text{pot}}$, which act non-trivially on only two plaquettes, one carrying a clockwise and the other carrying a anticlockwise flux as shown in (a). These two plaquettes are *flippable*. $\mathcal{O}_{\text{kin}}$ is an off-diagonal operator, while $\mathcal{O}_{\text{pot}}$ is a diagonal operator. Any of the other 14 possible states (two of which are illustrated in (b)) gets annihilated by both the operators. $\mathcal{O}_{\text{pot}}$ is a diagonal operator, it counts the total number of flippable plaquettes on the lattice.

The $U(1)$ quantum link model (QLM) and the quantum dimer model (QDM) that we will be concerned with in this paper use the quantum spin $S = \frac{1}{2}$ representation for the gauge fields, and thus a two-dimensional local Hilbert space. Since the $E_{\text{flux}}$ is trivial for this case, it can be omitted. The resulting model has a highly constraining kinetic term which acts non-trivially on only two out of $2^4 = 16$ possible states for the plaquette, corresponding to clockwise and anticlockwise circulation of electric flux (see Fig. 2). We call the corresponding plaquettes as *flippable*, while all other flux-configurations are non-flippable. This model studied in the context of high-energy physics [62] displays novel confined phases which spontaneously break the lattice translation symmetry and gives rise to half-integer flux tubes joining static charges, which repel each other. In the context of condensed matter physics, they have been used to represent the gauge invariant low-energy subspace of quantum spin-ice [61], and a spin-liquid on the pyrochlore lattice [60, 70]. To see such interesting physics however, one needs the additional operator

$$\mathcal{O}_{\text{pot}} = \sum_{\square} \left(U_{\square} + U_{\square}^{\dagger}\right)^2 = \sum_{\square}(U_{\square}U_{\square}^{\dagger} + U_{\square}^{\dagger}U_{\square}), \tag{3}$$

which counts the total number of flippable plaquettes on the lattice (see Fig. 2). The Hamiltonian of the $U(1)$ QLM we will be concerned with in this paper:

$$\mathcal{H}_{\text{QLM}} = \mathcal{O}_{\text{kin}} + \lambda \mathcal{O}_{\text{pot}}, \qquad G_{\text{r}}^{\text{QLM}}\,|\psi\rangle \;\; = (\nabla \cdot E)_{\text{r}}\,|\psi\rangle = 0. \tag{4}$$

The physical state $|\psi\rangle$ satisfies the constraint imposed by $G^{\text{QLM}}$ at all lattice sites, which is pictorially represented in Fig. 3. A very closely related model is the QDM, initially used by Rokhsar and Kivelson to realize the non-Néel phase of antiferromagnets as a potential route to high-temperature superconductivity [58], and hence these models are also called Rokhsar-Kivelson models. Subsequently they were also used to study resonating valence bond phases and fractionalized excitations [59]. The QDM has the same Hamiltonian as the QLM, but a different Gauss' Law:

$$\mathcal{H}_{\text{QDM}} = \mathcal{O}_{\text{kin}} + \lambda \mathcal{O}_{\text{pot}}, \qquad G_{\text{r}}^{\text{QDM}}\,|\chi\rangle \;\; = (\nabla \cdot E)_{\text{r}}\,|\chi\rangle = (-1)^{x_1 + x_2}\,|\chi\rangle. \tag{5}$$

Here $r = (x_1, x_2)$ denotes the lattice index. Since the Gauss law for the models are completely different, the physical states $|\psi\rangle$ and $|\chi\rangle$ for the two models do not overlap. Physically, the states selected by the QLM has zero charges on the vertex, while those selected by the QDM have staggered $\pm 1$ charges arranged throughout the lattice volume (see Fig 3). The charges in the QDM, however, are background charged without any associated dynamics. The latter can be interpreted as a non-relativistic one-form symmetry of the Abelian gauge theory [78]. The constraints are such that 6 states are allowed by the $G^{\text{QLM}}$ for each site, while 4 are allowed by $G^{\text{QDM}}$ at each site. In this sense, the QDM represents a more constrained model than the QLM. For notational efficiency, we will denote the Hamiltonian of both models as $\mathcal{H}$, and use the superscript on the Gauss Law for the respective models.

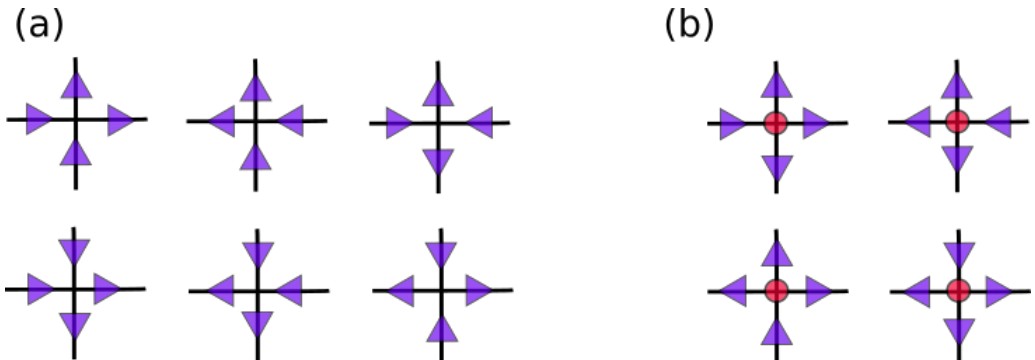

Figure 3: Gauss' laws for the QLM and the QDM. Six allowed states (a) for zero charge at the vertex as shown in panel (a), which is the case in the QLM, while for a staggered charge $Q = \pm 1$, a site has four states each for the unit positive and the unit negative charges. Panel (b) shows the states corresponding to $Q_r = 1$ at the vertex. The ones corresponding to the unit negative charge can obtained by flipping the individual links.

## 3 Global symmetries

Since we are interested in investigating the structure of anomalous eigenstates in the spectrum, we will discuss the different global symmetries of the two models which will be heavily exploited to facilitate the exact diagonalization (ED) studies. Besides the gauge symmetry, there is no other local symmetry in the system. However, there are a host of global symmetries, both discrete and continuous. It is important to note that not all these symmetries are mutually commuting.

The QLM has the following symmetries on a lattice of dimensions $L_x \times L_y$ with periodic boundary conditions in both directions:

- Lattice translations $T_i$, with $\hat{i} = x, y$ translates the operators one lattice spacing in the $x$ or the $y$ direction: ${}^{T_i}\mathcal{O}_{x,y} = \mathcal{O}_{x+\hat{i}, y+\hat{i}}$, where $\mathcal{O}_{x,y} = U_{x,y}, U^{\dagger}_{x,y}, E_{x,y}$.

- Charge conjugation $C$ is an internal symmetry which flips the electric flux of each link, ${}^{C}E_{xy} = -E_{xy}$, and conjugates the gauge field: ${}^{C}U_{xy} = U^{\dagger}_{xy}$ and ${}^{C}U^{\dagger}_{xy} = U_{xy}$.

- Reflection symmetry along the $x$ and $y$ axis aligned along the lattice axes.

- Lattice rotation symmetry involving $\pi/2$-rotations for $L_x = L_y$ and $\pi$-rotation for rectangular lattices for $L_x \neq L_y$.

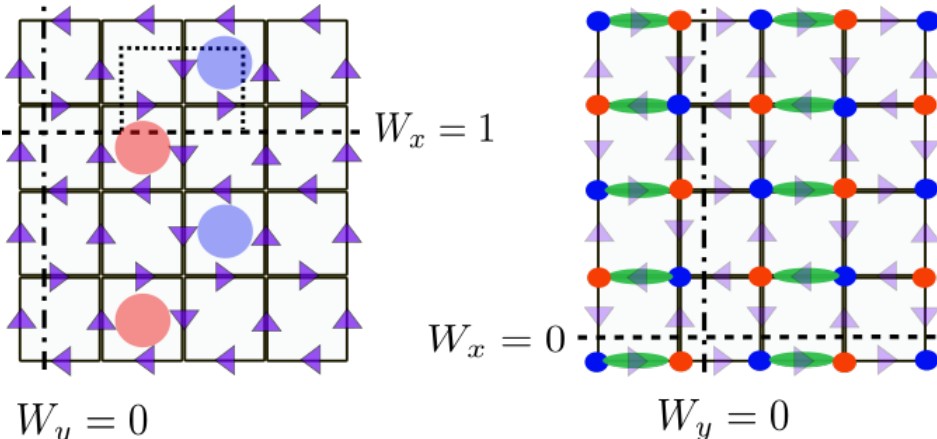

Figure 4: Examples of configurations of the QLM (left) and QDM (right) on $L_x = L_y = 4$ lattices with periodic boundary conditions in both directions, such that the leftmost and the rightmost vertical links are identical, and the topmost and the bottom-most horizonal links are identical. There is no net charge on the vertex for the QLM. The clockwise and the anti-clockwise flippable plaquettes are indicated by red and blue shaded circles respectively. The dashed and dot-dashed lines show the links which need to be summed to obtain the x- and y-winding respectively. Note that the dotted line indicates a deformation of the dashed line, and demonstrates the invariance of the windings under local deformations. The right panel shows an example configuration of the QDM in the so-called columnar phase. Note the staggered distribution of the static charges $\pm 1$, and the connection between the electric flux and dimers (shown as green ellipses), as explained in the text. The winding numbers are still good quantum numbers, but are insensitive only to deformations involving even number of lattice sites.

- Winding numbers $W_x, W_y$, as defined below generate global $U(1) \times U(1)$ symmetry:

$$W_x = \sum_{r=(x,y_0)} E_{x,y_0}, \qquad W_y = \sum_{r=(x_0,y)} E_{x_0,y}. \tag{6}$$

  The summation needs to be taken along the x-direction at $y = y_0$ for vertical links to compute $W_x$ and along the y-direction at $x = x_0$ for horizontal links to compute $W_y$, as shown in Fig. 4. Each winding sector is thus labeled by two integers $(W_x, W_y)$ and is topologically distinct from another sector with a different $(W_x, W_y)$.

We note that the Gauss law of the QLM preserves all the above symmetries. For the QDM, the situation is more subtle. Due to the Gauss law of the QDM, $G_r^{QDM}$, some global symmetries need to be reinterpreted:

- Lattice translation by two lattice spacings in either direction is still a symmetry due to the staggered background charge configuration. Shifts by a single lattice spacing is not a symmetry of $G^{QDM}$.

- Charge conjugation is no longer a symmetry of the QDM, since the background charge distribution on the lattice is staggered. Flipping the electric flux at each link results in a local inversion of the charge at the site, and the resulting state belongs to a different Hilbert space. However, it is possible to define operators for the combined action of charge conjugation with a single lattice translation: $\mathbb{T}_i = C T_i$. These operators are symmetries of both the Hamiltonian and $G_r^{QDM}$.

- It is possible to have $\pi/2$ rotations in the QDM on lattices with $L_x = L_y$ in two different ways, about each site or about the center of a plaquette. Since the latter changes the local charge at a site, one needs to compound this with a charge conjugation.

- Reflections about the lattice axes keep the charge distribution unchanged and are symmetries of the model.

- Even though one cannot define winding numbers in the presence of charges, for the QDM it is still possible to characterize basis states in terms of their winding numbers $(W_x, W_y)$, just as in the QLM case (see Fig. 4). This is due to the fact that the charge distribution is static and does not change with any parameter.

We further note that it is possible to rewrite the QDM in terms of the dimer variables $D_{xy}$ using the following mapping with the electric fluxes: $E_{xy} = (-1)^{x_1+x_2}(D_{xy} - \frac{1}{2})$, together with the constraint that every site is touched only by a single dimer. For even-parity sites $((-1)^{x_1+x_2} = 1)$, $E_{xy} = \frac{1}{2}$ corresponds to the presence of a dimer on the bond xy $(D_{xy} = 1)$, while $E_{xy} = -\frac{1}{2}$ indicates its absence $(D_{xy} = 0)$. For odd sites, this relation is reversed. Fig 4 shows a QDM configuration decorated with both electric flux and dimers.

For the rest of the paper, we will refer the Hamiltonian for both the QLM and the QDM as $\mathcal{H}$ since they are identical. When the distinction is required, we will use appropriate subscripts.

### 3.1 Index theorem at $\lambda = 0$

Additionally, at $\lambda = 0$, the Hamiltonian $\mathcal{H}$ anticommutes with the operator: $\mathbb{C} = \prod_{xy} E_{xy}$ where only the horizontal (vertical) links on even y (x) contribute to the product such that each elementary plaquette contains only one such link. As a consequence, for any eigenstate with energy $E \neq 0$, there is another eigenstate with energy $-E$: $(\mathbb{C}|E\rangle = |-E\rangle)$. Both $\mathcal{H}$ and $\mathbb{C}$ commute with a space reflection symmetry defined either along the horizontal or the vertical axis dividing the lattice in two equal halves. Remarkably, any Hamiltonian with these properties has exact zero-energy eigenstates whose number scales exponentially in the system size due to an index theorem shown in Ref. [55]. These zero modes are the only eigenstates of $\mathcal{O}_{kin}$ that have a well-defined "chiral charge" of $\pm 1$ under the action of $\mathbb{C}$. Furthermore, the index theorem ensures that these modes do not mix with the nonzero modes of $\mathcal{O}_{kin}$ in spite of the exponentially small level spacing in system size. This index theorem is violated for nonzero $\lambda$ because $\{\mathcal{H}, \mathbb{C}\} \neq 0$ when $\lambda \neq 0$ and it is then expected that the zero modes and the nonzero modes of $\mathcal{O}_{kin}$ hybridize with each other to give the new eigenstates. For the QLM and the QDM, the winding numbers $(W_x, W_y)$ additionally label topologically disconnected sectors and these protected zero modes are present in each winding number sector when $\lambda = 0$. We provide a numerical evidence of this for the QLM with $L_x = 8, L_y = 4$ in Fig. 5 where the many-body density of states, $\rho(E)$, after diagonalizing $\mathcal{O}_{kin}$ is shown for four winding number sectors, $(W_x, W_y) = (0, 0), (0, 1), (1, 0), (1, 1)$, respectively.

## 4 Methods: Exact Diagonalization

In order to study the properties of the eigenstates of $\mathcal{H}$ in the appropriate Gauss law sectors for the QLM or the QDM, we use large-scale exact diagonalization (ED). The complexity of the problem scales exponentially in the system-size, even after the projection into Gauss' law sectors. However, with the appropriate use of global symmetries, one can access larger system sizes. In our study, we have used the winding number symmetry to block diagonalize the Hamiltonian into different sectors characterized by integers $(W_x, W_y)$. In addition, the translation invariance is exploited to further reduce the size of the system being diagonalized (see

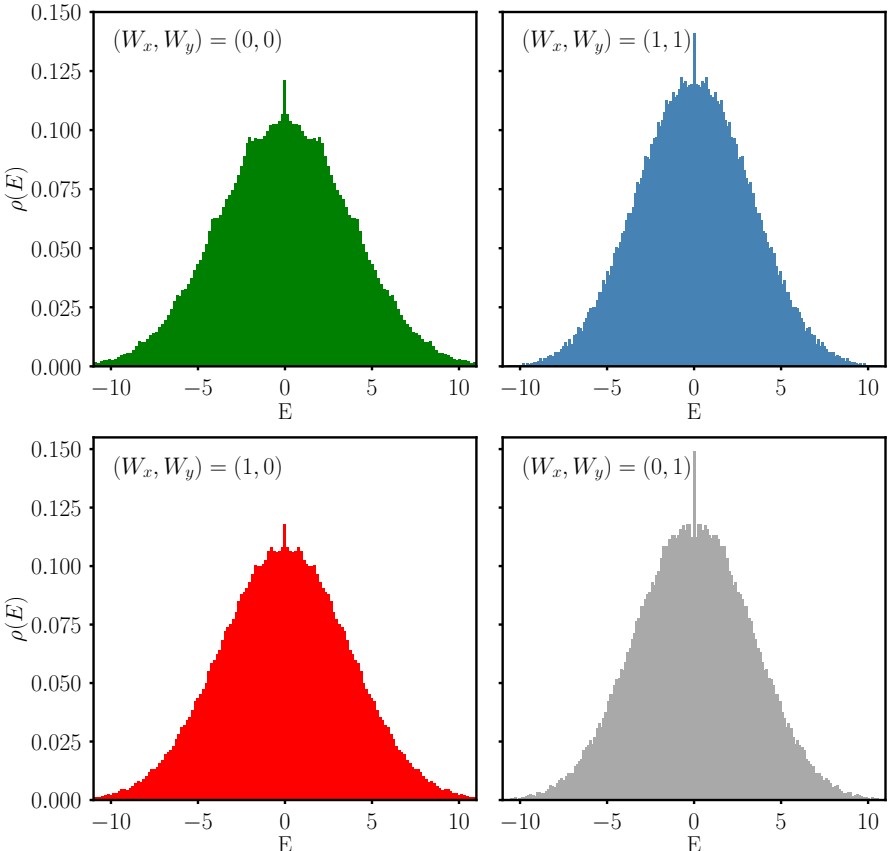

Figure 5: The many-body density of states, $\rho(E)$, obtained using exact diagonalization for the QLM with $L_x = 8, L_y = 4$, for the winding number sectors $(W_x, W_y) = (0,0)$ (top left), $(W_x, W_y) = (1,1)$ (top right), $(W_x, W_y) = (1,0)$ (bottom left) and $(W_x, W_y) = (0,1)$ (bottom right) at the coupling $\lambda = 0$. The Hilbert space dimensions are 1159166, 95760, 662944, and 152660 respectively, while the number of zero modes are 5312, 472, 2532, and 1234 respectively.

Ref. [79] for a lucid exposition on the technical aspects). The use of each of these symmetries decrease the size of the Hamiltonian approximately by a factor proportional to the volume of the system. Additionally, one can account for commuting discrete symmetries (such as the charge conjugation for QLM, the reflection symmetry) but that only decreases the Hamiltonian size by factors of 2. We have only implemented this in certain cases, especially when we study the level spacing distribution of the eigenenergies. Note that QDM has the more restrictive Gauss Law constraint than the QLM for a fixed lattice size. We list the allowed states in the $(k_x, k_y) = (0,0)$; $(W_x, W_y) = (0,0)$ sector for different system sizes of the QLM in Table 1 and of the QDM in Table 2. We used Intel MKL and Scipy routines to diagonalize matrices with dimensions of approximately $D \approx 50000$.

One important question we can immediately address with the eigenvalues obtained from ED is the issue of integrability. While it is conventional wisdom to claim the non-integrability of pure gauge theories in $(2+1)$-d, it is also instructive to demonstrate this directly using the method discussed in Ref. [80]. Here, one constructs the distribution of consecutive level spacing ratios $\tilde{r}$ (with support in $[0,1]$) of the Hamiltonian after projecting it to a sector with all commuting symmetries resolved. The level spacing ratios, $\tilde{r}$ are defined as

$$\tilde{r} = \min\left\{r_n, \frac{1}{r_n}\right\} \leq 1, \qquad r_n = \frac{s_n}{s_{n-1}}, \qquad s_n = E_{n+1} - E_n. \tag{7}$$

Table 1: Hilbert space dimension of the $(k_x, k_y) = (0, 0)$ sector in the largest winding number sector $(W_x, W_y) = (0, 0)$ of the QLM. The total number of states in any other sector will be smaller than this.

| Hilbert space in Quantum Link Model | | | |
|---|---|---|---|
| $(L_x, L_y)$ | Gauss law | $(W_x, W_y) = (0, 0)$ | $(k_x, k_y) = (0, 0)$ |
| (8, 2) | 7074 | 2214 | 142 |
| (10, 2) | 61098 | 17906 | 902 |
| (12, 2) | 539634 | 147578 | 6166 |
| (14, 2) | 4815738 | 1232454 | 44046 |
| (16, 2) | 43177794 | 10393254 | 324862 |
| (4, 4) | 2970 | 990 | 70 |
| (6, 4) | 98466 | 32810 | 1384 |
| (8, 4) | 3500970 | 1159166 | 36360 |
| (6, 6) | 16448400 | 5482716 | 152416 |

Table 2: Hilbert space dimension of the $(k_x, k_y) = (0, 0)$ sector in the largest winding number sector $(W_x, W_y) = (0, 0)$ of the QDM. The total number of states in any other sector will be smaller than this.

| Hilbert space in Quantum Dimer Model | | | |
|---|---|---|---|
| $(L_x, L_y)$ | Gauss law | $(W_x, W_y) = (0, 0)$ | $(k_x, k_y) = (0, 0)$ |
| (8, 2) | 1156 | 384 | 29 |
| (10, 2) | 6728 | 2004 | 106 |
| (12, 2) | 39204 | 10672 | 460 |
| (14, 2) | 228488 | 57628 | 2077 |
| (6, 4) | 3108 | 1456 | 71 |
| (8, 4) | 39952 | 17412 | 571 |
| (10, 4) | 537636 | 216016 | 5490 |
| (12, 4) | 7379216 | 2739588 | 57379 |
| (6, 6) | 90176 | 44176 | 1256 |
| (8, 6) | 3113860 | 1504896 | 31464 |

When the model is non-integrable, one expects a Gaussian orthogonal ensemble (GOE) distribution, while a Poisson distribution is expected for an integrable system [81]:

$$P_{\text{GOE}}(\tilde{r}) = \frac{27}{4} \frac{\tilde{r} + \tilde{r}^2}{(1 + \tilde{r} + \tilde{r}^2)^{5/2}} \; ; \qquad P_{\text{P}}(\tilde{r}) = \frac{2}{(1 + \tilde{r})^2} \; . \qquad (8)$$

after resolving all the mutually commuting symmetries of the Hamiltonian.

$P(\tilde{r})$ for the QLM and the QDM strongly indicates that the models are non-integrable when $|\lambda| \lesssim \mathcal{O}(1)$, as shown in Figure 6. The plots show $P(\tilde{r})$ in the $(0, 0)$ winding sectors of QLM (left) and QDM (right). For QLM on the lattice $(L_x, L_y) = (12, 2)$, the Hamiltonian (of dimension 3068) has been diagonalized in the sector with quantum numbers $(k_x, k_y, C, S_x) = (\pi/6, \pi, +1, -1)$, where $S_x$ is the reflection about the x-axis. The results follow $P_{\text{GOE}}$ for both small and large $\lambda$ values as shown on Figure 6 (left). This result was cross-checked on the larger lattice $(8, 4)$ with the Hamiltonian (dimension 18084) diagonalized in the sector with quantum numbers $(k_x, k_y, C) = (\pi/4, \pi/2, -1)$.

A similar check on the QDM yields a curious observation, namely, that while for small-$\lambda$ the distribution of level spacing ratios agree with that of $P_{\text{GOE}}(\tilde{r})$, for large values of $\lambda$, there is considerable deviation from this form towards $P_{\text{P}}(\tilde{r})$. We used three different lattices to verify

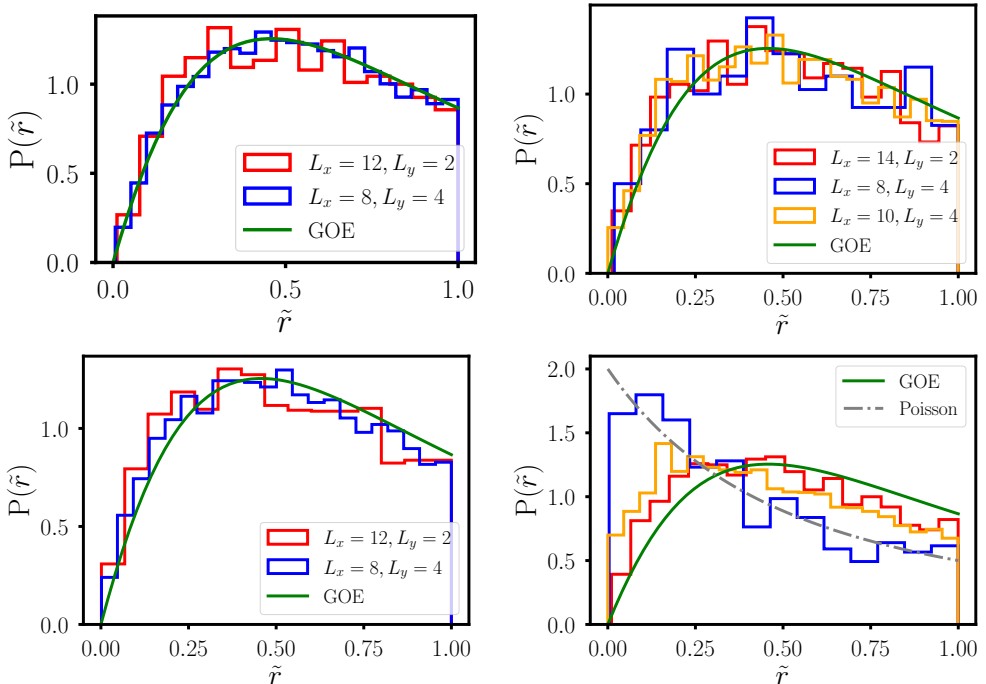

Figure 6: Level spacing ratio distribution $P(\tilde{r})$ vs $\tilde{r}$ for the QLM (left) and QDM (right) for different lattice sizes and different couplings, $\lambda = 0.1$ (above) and $\lambda = 3.13$ (below). The QLM results were obtained on two different lattices of size $(12, 2)$ and $(8, 4)$ in a specified symmetry sector with 3068 and 18048 states respectively. To make the histogram for the two cases, the number of bins used are 15, 22, respectively. For the QDM, we have used three different lattices $(14, 2), (8, 4)$ and $(10, 4)$ in symmetry sectors with Hilbert space dimensions of 2041, 532, and 5350 respectively. To make the histograms 18, 13, 22 bins were used respectively. The histograms indicate the non-integrability of the models for $\mathcal{O}(\lambda) \sim 1$.

this. The lattice $(14, 2)$ was diagonalized in the sector with quantum numbers $(k_x, k_y, S_y) = (\pi/7, \pi, -1)$ having dimension 2041, the lattice $(8, 4)$ in the sector $(k_x, k_y) = (\pi/2, \pi/2)$ with dimension 532, and the lattice $(10, 4)$ in the sector $(k_x, k_y) = (\pi/5, \pi/2)$ having dimension 5350. For the QDM, $S_x$ is again the reflection symmetry about the x-axis, while $(k_x, k_y)$ is the symmetry operation corresponding to $(\mathbb{T}_x, \mathbb{T}_y)$. For large negative $\lambda$, the deviation from $P_{\text{GOE}}(\tilde{r})$ is progressively pronounced for the bigger lattices. This behavior of the spectrum of the QDM is consistent with the one reported in [82] where strong deviations from the ETH were seen on rectangular lattices when $\mathcal{O}_{\text{pot}}$ dominated over $\mathcal{O}_{\text{kin}}$. Interestingly, the QLM does not seem to show any such deviations in the same regime from the level statistics data shown in Fig. 6. The possibility of disorder-free localization in QLMs as discussed in Ref. [83] is not relevant here since we restrict ourselves to one particular superselection sector defined by Eq. 4 and do not start with initial states that involve many superselection sectors.

Importantly, these figures clearly indicate that for small and moderate values of $\lambda$ that have been used in the subsequent sections, both models are clearly non-integrable, and are expected to follow predictions of ETH. The presence of any anomalous eigenstate here therefore points to the presence of quantum many-body scars in the system.

# 5 Diagnostics of the Quantum Many-Body Scars

While typical excited states are expected to be delocalized in the Hilbert space spanned by a simple unentangled basis, the special quantum scar states are distinguished by their localization in this particular basis. In our case, we work in the electric flux basis, which is the most natural local basis for these models. The distinguishing feature of the scars is that when the amplitude of scar wavefunction is plotted in this basis, one observes a very localized distribution instead of a distribution over all basis states. In addition, these eigenstates are characterized by an anomalous value of several observables detailed below.

- Bipartite entanglement entropy, $S_{L/2}$ for each energy eigenstate $|\Psi\rangle$,

$$S_{L/2} = -\text{Tr}[\rho_A \ln \rho_A], \quad \rho_A = \text{Tr}_{\overline{A}} |\Psi\rangle \langle\Psi|, \tag{9}$$

  where $\rho_A$ is the reduced density matrix obtained by partitioning the system into two equal parts $A$ and $\overline{A}$. While we have mostly considered bipartitions along the y-direction for our results, certain results in Sec. 8 use cuts both along the x- and y-direction. Fig 16 displays the bipartition axes. Technical details about the computation of $S_{L/2}$ can be found in the Supplementary Material of [42].

- Anomalous localization of the scar wavefunction $|\psi_S\rangle$ in the flux basis is one of the best ways of identifying the scar state. For certain instances (particularly in the QDM, with $L_y = 4$ and $L_x = 6, 8$), there are several such states at the same energy, distinct from the ETH band in terms of $S_{L/2}$. In these, we have found it useful to implement an entanglement-minimization procedure to create eigenstates, which are simpler than naively obtained from the ED routine. The algorithm is explained in the Appendix A.

- The Shannon entropy, $S_1$, defined as

$$S_1 = -\sum_\alpha |\psi_\alpha|^2 \ln |\psi_\alpha|^2; \quad |\Psi\rangle = \sum_{\alpha=1}^{\mathcal{N}} \psi_\alpha |\alpha\rangle, \tag{10}$$

  when the eigenstate is expressed in a given basis $|\alpha\rangle$ with $\mathcal{N}$ basis states.

- The correlation of the summed electric flux operators, defined as

$$E_{\text{corr}} = \frac{1}{L_x} \sum_x \langle E_{\hat{j}}(x) E_{\hat{j}}(x + \hat{i}) \rangle, \tag{11}$$

  where $E_{\hat{j}}(x_0) = \sum_y E_{\mathbf{r}=(x_0,y)}$ equals the sum of the electric fluxes along all the y-links for a given $x_0$ thus acting as a smeared electric flux operator. Since $E_{\text{corr}}$ reduces to the sum of local correlation functions, it is expected to attain values that approach the ETH prediction for typical high-energy eigenstates.

## 5.1 Classifying scars using zero and nonzero modes of $\mathcal{O}_{\text{kin}}$

The anomalous eigenstates observed in the QLM and the QDM with $\mathcal{H}$ are always typically localized in the Hilbert space, and can be classified under the following categories using the zero and the nonzero modes of $\mathcal{O}_{\text{kin}}$:

- Type-I: These are the first ones to be observed in the $U(1)$ QLM as shown in Ref. [42], where the eigenstate $|\Psi\rangle$ of the Hamiltonian is a simultaneous eigenstate of $(\mathcal{O}_{\text{kin}}, \mathcal{O}_{\text{pot}})$, with the eigenvalue of $\mathcal{O}_{\text{kin}}$ to be 0. As a consequence of the index theorem shown in

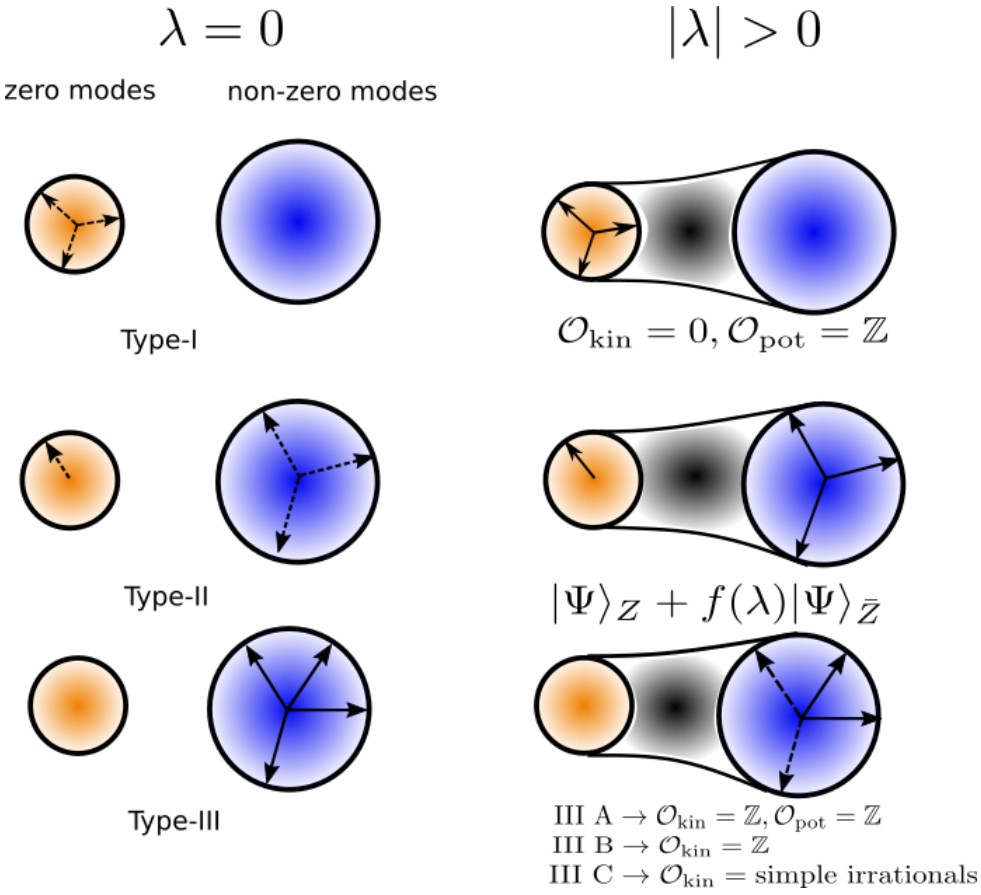

Figure 7: Schematic representation of the different varieties of quantum scars discussed here in terms of the zero and the nonzero modes of $\mathcal{O}_{\mathrm{kin}}$. The arrows indicate how the scars are embedded in the spectrum for different values of $\lambda$. Type-I scars are eigenstates of $\mathcal{H}$ for all $\lambda$, and are embedded in the nullspace (dashed arrow), and only exhibit anomalous behaviour (solid arrows) in presence of $\mathcal{O}_{\mathrm{pot}}$. Similarly, type-IIIA scars are eigenstates of $\mathcal{H}$ for all $\lambda$ (solid arrows), type-II scars are eigenstates for $\lambda \neq 0$ (hence they are denoted by dashed arrows for $\lambda = 0$ and solid arrows otherwise) while type-IIIB and type-IIIC scars are eigenstates for $\lambda = 0$ (where they are indicated with solid arrows). The black shading indicates hybridization of the zero and the non-zero modes of $\mathcal{O}_{\mathrm{kin}}$ for $\lambda \neq 0$. The total number of arrows in the plot is schematic.

Ref. [55], there is an exponential number of zero modes of $\mathcal{O}_{\mathrm{kin}}$ which are expected to satisfy the ETH since these are mid-spectrum states. A generic linear combination of such zero modes is also expected to be locally thermal. However, in certain cases, these zero modes overlap in a pseudo-random fashion to produce eigenstates of $\mathcal{O}_{\mathrm{pot}}$ with an integer eigenvalue $\mathbb{Z}_{\mathrm{pot}}$, indicated with solid arrows in Fig. 7 (top panel). Such anomalous zero modes clearly violate the ETH [42] since these states stay unchanged at any $\lambda$ being simultaneous eigenstates of $\mathcal{O}_{\mathrm{kin}}$ and $\mathcal{O}_{\mathrm{pot}}$ in spite of the exponentially small level spacing around them. As a result, these anomalous zero modes turn out to be much more localized in the Hilbert space than any typical zero mode of $\mathcal{O}_{\mathrm{kin}}$. However, at $\lambda = 0$, the states mix with the other states in the nullspace, and do not give any dynamical signatures. This is indicated with dotted lines in the top panel of Fig. 7.

- Type-II: For this class of scars, the anomalous eigenstates are not individual eigenstates of either $\mathcal{O}_{\text{kin}}$ or $\mathcal{O}_{\text{pot}}$. For such scars, $\langle \mathcal{O}_{\text{kin}} \rangle = 0$ and $\langle \mathcal{O}_{\text{pot}} \rangle = \mathbb{Z}_{\text{pot}}$ which does not change with $\lambda$. Thus, the total energy of type-II scars equals $\lambda \mathbb{Z}_{\text{pot}}$ exactly like the type-I scars. However, unlike the type-I scars, these scars are composed of both the zero and nonzero modes of $\mathcal{O}_{\text{kin}}$ (hence the dotted lines in the middle left panel in Fig. 7) with a very specific mixing between certain basis states of the zero and the nonzero mode subspaces of $\mathcal{O}_{\text{kin}}$. In particular, these (unnormalized) scars have the form: $|\Psi\rangle = |\Psi\rangle_Z + f(\lambda)|\Psi\rangle_{\bar{Z}}$, where the subscript $Z(\bar{Z})$ denotes the zero (nonzero) mode subspace of $\mathcal{O}_{\text{kin}}$. Note that $|\Psi\rangle_Z$ and $|\Psi\rangle_{\bar{Z}}$ remain independent of $\lambda$ with only their relative phase, $f(\lambda)$, changing with $\lambda$ implying that such eigenstates violate the ETH. These states require $\lambda \neq 0$ to be eigenstates of the Hamiltonian $\mathcal{H}$, since a mixing of the zero and the nonzero modes of $\mathcal{O}_{\text{kin}}$ is necessary and hence the solid lines in the middle right panel of Fig. 7). It is useful to point out here that anomalous eigenstates with a somewhat similar structure were previously found in a model with Hilbert space fragmentation [84] but the appearance of these type-II scars formed out of both the zero and nonzero modes of $\mathcal{O}_{\text{kin}}$ does not require any form of fragmentation.

- Type-III: These scars are characterized by eigenstates of $\mathcal{O}_{\text{kin}}$ where the eigenvalue is either an integer different from 0, or a simple irrational number such that a suitable power gives back an integer. These nonzero integers or simple irrational numbers are typically $O(1)$ and do not scale with system size.

  Since $\mathcal{H}$ $(\lambda = 0) = \mathcal{O}_{\text{kin}}$ represents a strongly interacting non-integrable model, typical high-energy eigenstates are expected to have complicated irrational energy eigenvalues that cannot be written in any closed form. The only way special eigenvalues $E$ like the ones described in the previous paragraph can be obtained is to have a factorization of the corresponding characteristic equation $\det(\mathcal{O}_{\text{kin}} - \mathbb{I}E) = 0$ (where $\mathbb{I}$ represents an identity matrix of the same dimension as $\mathcal{O}_{\text{kin}}$) in the form $(E - \mathbb{Z}_{\text{kin}})^{n_0}(E + \mathbb{Z}_{\text{kin}})^{n_0} E^{n_z} f_0(E) = 0$ or $(E^{2n_1} - \mathbb{Z}_{\text{kin}})^{n_2} E^{n_z} f_2(E) = 0$ where $f_{0,2}(E)$ represents a high-degree polynomial in $E$ which, in general, has complicated irrational numbers as roots. The former factorization gives $n_0$ degenerate eigenvalues with $\pm \mathbb{Z}_{\text{kin}}$ while the latter factorization gives $n_1 n_2$ eigenvalues $\pm(\mathbb{Z}_{\text{kin}})^{1/(2n_1)}$ where $\mathbb{Z}_{\text{kin}}, n_{0,1,2,z}$ are all integers. These roots always come in pairs since at $\lambda = 0$, the spectrum is symmetric around $E = 0$. Such special factorizations of the characteristic equation imply that the corresponding eigenvectors are atypical. This argument does not, however, work for the $n_z$ zero modes of $\mathcal{O}_{\text{kin}}$ since both their degeneracy and their energy $E = 0$ arises due to the index theorem of Ref. [55] and the factorization $E^{n_z}$ of the characteristic polynomial is simply its consequence.

  We can further subdivide the type-III scars into three subclasses:

  - Type-IIIA: Both the eigenvalues of $\mathcal{O}_{\text{kin}}$ and $\mathcal{O}_{\text{pot}}$ are integers, but for $\mathcal{O}_{\text{kin}}$ it is an integer different from 0. These states are eigenstates of the Hamiltonian for any $\lambda$.

  - Type-IIIB: $\mathcal{O}_{\text{kin}}$ has a nonzero integer eigenvalue, but $\mathcal{O}_{\text{pot}}$ does not have any definite eigenvalue. Consequently, these are only eigenstates of the entire Hamiltonian at $\lambda = 0$.

  - Type-IIIC: $\mathcal{O}_{\text{kin}}$ has a simple irrational number as an eigenvalue (such as $\pm\sqrt{2}$), but $\mathcal{O}_{\text{pot}}$ does not have any definite eigenvalue. Consequently, these are only eigenstates of the entire Hamiltonian at $\lambda = 0$.

The solid and dashed lines in the (nonzero) zero mode space in bottom panel of Fig. 7) indicate these possibilities. Scars with such characteristic energies (non-zero integers and simple irrationals) have been reported earlier in other non-integrable models like the

one-dimensional and the two-dimensional PXP models [13, 85] and a one-dimensional PXP-like model with a larger blockade radius [86]. Interestingly, all these models satisfy the aforementioned index theorem. However, unlike the scars in the previous examples, the type-IIIA scars being reported here also survive as eigenstates at any finite $\lambda$ where the index theorem is no longer valid.

# 6 Extraction of scars from the spectrum

Based on previous experience [42], we anticipate simultaneous eigenstates of $\mathcal{O}_{\text{kin}}$ and $\mathcal{O}_{\text{pot}}$ to behave like quantum many-body scars, as they deform smoothly with the $\lambda$, unlike typical ETH satisfying eigenstates. As explained before, there is an exponentially large eigenspace of zero modes of the $\mathcal{O}_{\text{kin}}$ operator. We determine linear combinations of the zero-modes which could be eigenstates of $\mathcal{O}_{\text{pot}}$, following the algorithm below:

- Consider the ordered eigenbasis of $\mathcal{O}_{\text{kin}}$: $\mathcal{B}_{\mathcal{O}_{\text{kin}}} = \{|v_1\rangle, |v_2\rangle, \cdots, |v_{n-m}\rangle, |z_1\rangle, \cdots, |z_m\rangle\}$, where $|z_i\rangle$ are the zero-modes, and $|v_i\rangle$ are the remaining non-zero modes, which together span the full winding sector of dimension $n$. Expressed in this ordered basis, the $\mathcal{O}_{\text{pot}}$ operator is:

$$\mathcal{O}_{\text{pot}} = \left[ \begin{array}{c|c} [\mathbb{A}]_{(n-m)\times(n-m)} & [\mathbb{B}]_{(n-m)\times m} \\ \hline [\mathbb{C}]_{m\times(n-m)} & [\mathbb{V}_z]_{m\times m} \end{array} \right]_{n\times n}, \tag{12}$$

where the blocks denote: $(\mathbb{V}_z)_{ij} = \langle z_i|\mathcal{O}_{\text{pot}}|z_j\rangle$, $\mathbb{B}_{ij} = \langle v_i|\mathcal{O}_{\text{pot}}|z_j\rangle$, and so on.

- Any state residing solely in the zero-mode subspace, expressed in this basis looks like:

$$\left[ \begin{array}{c} \mathbb{O} \\ \hline \mathcal{C} \end{array} \right]_{n\times 1}, \tag{13}$$

where the upper block $\mathbb{O}$ is a column of $(n-m)$ zeros. The lower block $\mathcal{C}$ (of size $m \times 1$) are the components of the state along different zero-mode directions.

- For such a state to be an eigenstate of $\mathcal{O}_{\text{pot}}$, it must satisfy: $[\mathbb{V}_z] \times [\mathcal{C}] = V[\mathcal{C}]$, for some scalar $V$, and $[\mathbb{B}] \times [\mathcal{C}] = [0]$ (0 matrix of dimensionality $m \times 1$). We diagonalize the smaller block $\mathbb{V}_z$, and extract its orthonormal eigenstates with integer eigenvalues $V$ (say, $p$ in number), and denote this space as $\mathbb{P}_V = span\{|\psi_1\rangle, \ldots, |\psi_p\rangle\}$. These states could be potential scars, but need further investigation.

- The scars are those states in $\mathbb{P}_V$ which are annihilated by the block $\mathbb{B}$. We first project the domain of $\mathbb{B}$ onto $\mathbb{P}_V$ by determining the operator: $[\tilde{\mathbb{B}}]_{n-m\times p} = [\mathbb{B}]_{n-m\times m} \times [M]_{m\times p}$. The $i$'th column of $[M]$ represents the vector $\psi_i$ expressed in the zero-mode basis. The scar eigenvectors are those that form the the nullspace of $\tilde{\mathbb{B}}$, and can be conveniently obtained using the Singular Value Decomposition (SVD). The right-singular vectors corresponding to the vanishing singular values are our required scars.
  Note that these eigenstates so obtained, are expressed in terms of the $\psi_i$'s of $\mathbb{P}_V$, hence appropriate basis transformation is needed to re-express them in the flux basis.

Note further that the zero-mode subspace can be conveniently replaced by any other exactly degenerate eigenspace of $\mathcal{O}_{\text{kin}}$ to determine scars residing inside the same. In that case, only the ordered basis $\mathcal{B}_{\mathcal{O}_{\text{kin}}}$ have to be rearranged, all the subsequent steps will remain valid. This same procedure, therefore, works to identify both type-I and type-IIIA scars. Type-IIIB and type-IIIC scars can be identified straightforwardly by inspection of the eigenspectrum of

$\mathcal{O}_{\text{kin}}$ to search for eigenvalues that are non-zero integers or irrational numbers which square to an integer (up to machine precision). Identification of type-II scars is considerably more subtle: after the aforementioned types are classified, the remaining anomalous states (with low $S_{L/2}$ and total energy $\lambda\mathbb{Z}_{\text{pot}}$) are decomposed in eigenbasis of $\mathcal{O}_{\text{kin}}$ after which the amplitudes are rescaled separately such that the amplitude of a particular zero (nonzero) mode is chosen to stay unchanged with $\lambda$. The amplitudes of the other zero (nonzero) modes are rescaled accordingly. If all these amplitudes stay invariant with $\lambda$ after this procedure, then such eigenstates are classified as type-II scars.

# 7 Scars in the zero winding sectors of the QLM and the QDM

In this section, we consider the many-body spectrum of the QLM and the QDM for the largest topological sector with winding $(W_{\text{x}}, W_{\text{y}}) = (0,0)$ and show the presence of the different varieties of quantum many-body scars. For ease of presentation, we summarize the numerical results for the anomalous eigenstates based on ED in Table 3.

Table 3: A summary of the different varieties of scars obtained for different lattice dimensions $(L_{\text{x}}, L_{\text{y}})$ ($N_p = L_{\text{x}}L_{\text{y}}$ is the total number of plaquettes) for both the QLM and the QDM in the winding number sector $(W_{\text{x}}, W_{\text{y}}) = (0,0)$ based on ED calculations. Type indicates whether a scar is type I, II, IIIA, IIIB, IIIC as defined in Sec. 5.1, Degeneracy refers to the number of such scars with the same energy eigenvalue, $(\mathcal{O}_{\text{kin}}, \mathcal{O}_{\text{pot}})$ refers to the eigenvalues of these operators in the scar state with $\cdots$ indicating that the corresponding eigenvalue of $\mathcal{O}_{\text{kin}}$ ($\mathcal{O}_{\text{pot}}$) is not defined.

| $(L_{\text{x}}, L_{\text{y}})$ | Type | Degeneracy | $(\mathcal{O}_{\text{kin}}, \mathcal{O}_{\text{pot}})$ |
|---|---|---|---|
| Scars in QLM at $(W_x, W_y) = (0,0)$ | | | |
| $(L, 2)$ | Type I | 4 | $(0, N_p/2)$ |
| $(4,4)$ | Type I | 26 | $(0, 8)$ |
| | Type I | 12 | $(0, 6)$ |
| | Type IIIA | 6 | $(\pm 2, 8)$ |
| | Type IIIB | 12 | $(\pm 2, \cdots)$ |
| $(6,4)$ | Type I | 46 | $(0, 12)$ |
| | Type I | 8 | $(0, 10)$ |
| | Type II | 4 | $(\cdots, \cdots)$ |
| | Type IIIA | 2 | $(\pm 2, 12)$ |
| | Type IIIB | 5 | $(\pm 2, \cdots)$ |
| $(8,4)$ | Type I | 106 | $(0, 16)$ |
| | Type I | 12 | $(0, 14)$ |
| | Type IIIA | 2 | $(\pm 2, 16)$ |
| | Type IIIB | 1 | $(\pm 2, \cdots)$ |
| Scars in QDM at $(W_x, W_y) = (0,0)$ | | | |
| $(4,4)$ | Type I | 9 | $(0, 4)$ |
| | Type I | 1 | $(0, 6)$ |
| $(6,4)$ | Type I | 6 | $(0, 4)$ |
| $(8,4)$ | Type I | 4 | $(0, 8)$ |
| | Type I | 16 | $(0, 7)$ |
| | Type I | 8 | $(0, 4)$ |
| | Type IIIC | 16 | $(\pm\sqrt{2}, \cdots)$ |

## 7.1 Results for the QLM

**Type-I scars**: For systems of dimension $(L_x, L_y) = (L, 2)$, using ED for $8 \leq L \leq 14$, Ref. [42] showed the presence of 4 type-I scars, 1 each at momenta $(k_x, k_y) = (0,0), (\pi, \pi), (0, \pi), (\pi, 0)$ such that these states are also eigenstates of $(\mathcal{O}_{\text{kin}}, \mathcal{O}_{\text{pot}})$ with eigenvalue $(0, N_p/2)$ where $N_p = L_x L_y$ is the number of elementary plaquettes on the lattice. Expressing these quantum scars in terms of the basis states at the corresponding momentum shows that unlike in the neighboring mid-spectrum eigenstates, only a few of the basis states have non-zero coefficients for the scars which is a tell-tale signature of localization in the Hilbert space. For example, only 18 out of 44046 basis states at $(k_x, k_y) = (0,0)$ contribute to the formation of the scar for a system of dimension $(14, 2)$ [42].

ED results on wider systems of width $L_y = 4$ show several new features which are absent at $L_y = 2$. Let us first focus on the type-I scars. As summarized in Table 3, for a fixed $L_y = 4$, the degeneracy of type-I scars that are eigenstates of $(\mathcal{O}_{\text{kin}}, \mathcal{O}_{\text{pot}})$ with eigenvalues $(0, N_p/2)$ rapidly increase with increasing $L_x$: 26 such scars for $L_x = 4$, 46 such scars for $L_x = 6$ and 106 such scars for $L_x = 8$ suggesting a divergence of the number of such scars for $L_x \gg 1$ when $L_y = 4$. This is completely unlike the case with $L_y = 2$ where this number stays fixed to 4 with increasing $L_x$ based on the numerical evidence. Furthermore, new type-I scars with an eigenvalue of $(\mathcal{O}_{\text{kin}}, \mathcal{O}_{\text{pot}})$ equal to $(0, (L_x - 1)N_p/(2L_x))$ also appear for the wider ladders with the degeneracy being 12 for $L_x = 4, 8$, and 8 for $L_x = 6$ (Table 3).

Unlike in $L_y = 2$, the type-I scars for wider ladders with $L_y = 4$ can have momenta different from $(k_x, k_y) = (0,0), (\pi, \pi), (0, \pi), (\pi, 0)$. To illustrate this, the degeneracy of the type-I scars with eigenvalue of $(0, N_p/2)$ for $(\mathcal{O}_{\text{kin}}, \mathcal{O}_{\text{pot}})$ and their corresponding momenta $(k_x, k_y)$ for a system with dimension $(8, 4)$ are as follows: 1 for $(\pi, 3\pi/2)$, 1 for $(0, 3\pi/2)$, 5 for $(7\pi/4, \pi)$, 6 for $(3\pi/2, \pi)$, 5 for $(5\pi/4, \pi)$, 10 for $(\pi, \pi)$, 5 for $(3\pi/4, \pi)$, 6 for $(\pi/2, \pi)$, 5 for $(\pi/4, \pi)$, 9 for $(0, \pi)$, 1 for $(\pi, \pi/2)$, 1 for $(0, \pi/2)$, 5 for $(7\pi/4, 0)$, 6 for $(3\pi/2, 0)$, 5 for $(5\pi/4, 0)$, 9 for $(\pi, 0)$, 5 for $(3\pi/4, 0)$, 6 for $(\pi/2, 0)$, 5 for $(\pi/4, 0)$ and 10 for $(0, 0)$. For the same system dimension, there are 6 scars each at momenta $(0, \pi)$ and $(\pi, 0)$ with eigenvalues $(0, 14)$ for $(\mathcal{O}_{\text{kin}}, \mathcal{O}_{\text{pot}})$. Such type-I scars show up as clear outliers in the momentum-resolved data for both the Shannon entropy, $S_1$ (Eq. 10), and the electric flux correlator, $E_{\text{corr}}$ (Eq. 11), at finite $\lambda$ as shown in Fig. 8 for a specific momentum $(k_x, k_y) = (0, 3\pi/2)$ for the coupling $\lambda = -1.1$.

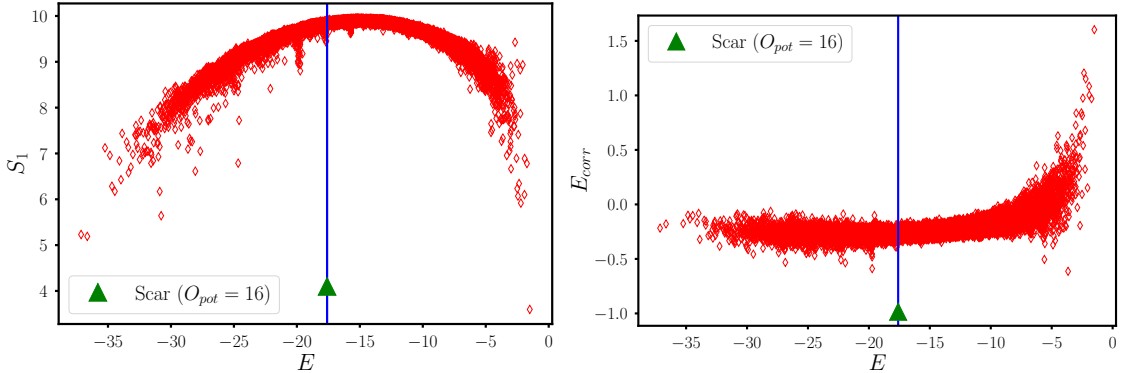

Figure 8: The Shannon entropy (left panel) and the electric flux correlator (right panel) shown for all the eigenstates at momentum $(k_x, k_y) = (0, 3\pi/2)$ for the QLM defined on a system of dimension $(L_x, L_y) = (8, 4)$ at a coupling $\lambda = -1.1$ in the winding number sector $(W_x, W_y) = (0, 0)$. The single type-I scar at this momentum with $(\mathcal{O}_{\text{kin}}, \mathcal{O}_{\text{pot}}) = (0, 16)$ and energy $E = 16\lambda$ (vertical blue line in both panels) shows up as an outlier (indicated by the green triangle) in both these quantities.

The localization of these type-I scars in the Hilbert space can be readily seen by expressing the amplitudes of any such scar in the basis states with momentum $(k_x, k_y)$. While a typical energy eigenstate at a nearby energy eigenvalue receives contributions from almost all the basis states in a pseudo-random manner, the scar states receive contributions from a very few number of basis states. For ease of visualization, we show the real-valued amplitudes for a scar with $(\mathcal{O}_{\text{kin}}, \mathcal{O}_{\text{pot}}) = (0, 16)$ at momentum $(k_x, k_y) = (0, 0)$ and another with $(\mathcal{O}_{\text{kin}}, \mathcal{O}_{\text{pot}}) = (0, 14)$ at momentum $(k_x, k_y) = (0, \pi)$ in Fig. 9 for a system with dimension $(8, 4)$ from which it is evident that most of the basis states (out of $\sim 36000$ basis states in each case) have no contribution within numerical precision to the many-body wavefunction of the quantum scars. Only 273 and 384 basis states contribute to the scar with amplitude greater than $10^{-10}$ in the two cases, respectively.

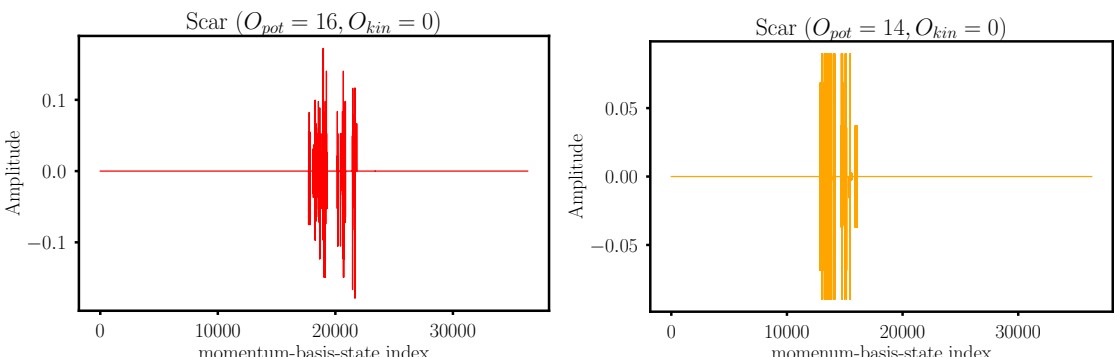

Figure 9: Amplitudes of two representative scar states expressed in their respective momentum basis for the $(8, 4)$ system at zero winding number. The different basis states are represented by integers on the x axis of both panels. The left (right) panel shows a scar with $(\mathcal{O}_{\text{kin}}, \mathcal{O}_{\text{pot}}) = (0, 16)$ $((0, 14))$ with momentum $(k_x, k_y) = (0, 0)$ $((0, \pi))$.

**Type-II & type-III scars**: Importantly, apart from the type-I scars present in systems with $L_{\text{y}} = 2$, other varieties of scars also emerge for the systems with $L_{\text{y}} = 4$ (Table 3). This is illustrated for the case of the system $(L_{\text{x}}, L_{\text{y}}) = (6, 4)$ in Fig. 10 both for $\lambda = 0$ and $\lambda \neq 0$ where the bipartite entanglement entropy, $S_{L/2}$ (Eq. 9), is shown for all the eigenstates in the zero winding number sector. At $\lambda = 0$ (top left panel of Fig. 10), we find 7 degenerate eigenstates with eigenvalues $\mathcal{O}_{\text{kin}} = +2$ and $\mathcal{O}_{\text{kin}} = -2$ respectively. These are anomalous eigenstates from the discussion of type-III scars in Sec. 5.1 and also show up as outliers in the bipartite entanglement entropy, $S_{L/2}$ (Eq. 9), as shown in Fig. 10 (top left panel).

Remarkably, 2 linear combinations of these scars with $\mathcal{O}_{\text{kin}} = +2$ $(-2)$ also diagonalize $\mathcal{O}_{\text{pot}}$ with eigenvalue 12 which result in type-IIIA scars that are eigenstates at any $\lambda$ with energy $E = 12\lambda + 2$ $(E = 12\lambda - 2)$ (Fig. 10, top right) whereas the other 5 scars are type-IIIB which are only present when $\lambda = 0$ and cease to be eigenstates of $\mathcal{H}$ for $\lambda \neq 0$. The localization of type-IIIA scars, which have momentum $(0, 0)$ and $(\pi, \pi)$ respectively, is evident by plotting the amplitude of one such scar with $(\mathcal{O}_{\text{kin}}, \mathcal{O}_{\text{pot}}) = (-2, 12)$ in the basis of states with $(k_x, k_y) = (0, 0)$ with only 32 out of the 1384 momentum states contributing to it as shown in Fig. 10 (bottom left). On the other hand, type-IIIB scars for the $(6, 4)$ system do not seem to have a well-defined momentum but expressing such a scar wavefunction directly in the basis of the electric flux Fock states again highlights their anomalous nature as well as the fact that these are not eigenstates of $\mathcal{O}_{\text{pot}}$ (Fig. 10, bottom right).

Apart from the 8 type-I scars with $E = 10\lambda$ at $\lambda \neq 0$, there are 4 other type-II scars which also have exactly the same energy eigenvalue of $E = 10\lambda$. However, as explained in

Sec. 5.1, these eigenstates are composed of a superposition of zero modes and nonzero modes of $\mathcal{O}_{kin}$ such that neither change as a function of $\lambda$, but the unnormalized wavefunction can be written such that only the relative phase factor between these modes changes with $\lambda$. The 4 type-II scars for the $(6,4)$ system have the quantum numbers $(k_x, k_y, C)$ equal to $(0, 0, +1)$, $(0, 0, -1)$, $(\pi, \pi, +1)$, and $(\pi, \pi, -1)$ respectively. The special form of such a quantum scar with $(k_x, k_y, C) = (0, 0, -1)$ is shown in Fig. 11 by expressing its wavefunction at two different nonzero values of $\lambda$ in terms of the eigenbasis of $\mathcal{O}_{kin}$ in order to demarcate the contributions from the zero and the nonzero modes. The amplitudes from the zero modes and the nonzero modes are separately normalized at the two different $\lambda$ by fixing only one particular amplitude along a chosen zero (nonzero) mode to be unchanged with $\lambda$. Remarkably, this makes the rescaled amplitudes equal at the two different couplings. The type-II scar is also extremely localized in the $\mathcal{O}_{kin}$ basis since it receives finite contributions from very few nonzero modes of $\lambda = 0$ (a linear superposition of the zero modes at $\lambda = 0$ can be reinterpreted as a single zero mode). This is completely different to what happens to a neighboring eigenstate at $\lambda \neq 0$ which is much more delocalized in the $\mathcal{O}_{kin}$ basis when contributions from the nonzero modes are considered. To the best of our knowledge, this is the first demonstration (albeit numerical) of such a peculiar mixing of the zero and nonzero modes of a many-body Hamiltonian ($\mathcal{O}_{kin}$) to stabilize type-II quantum scars in a non-integrable model when the index theorem is broken.

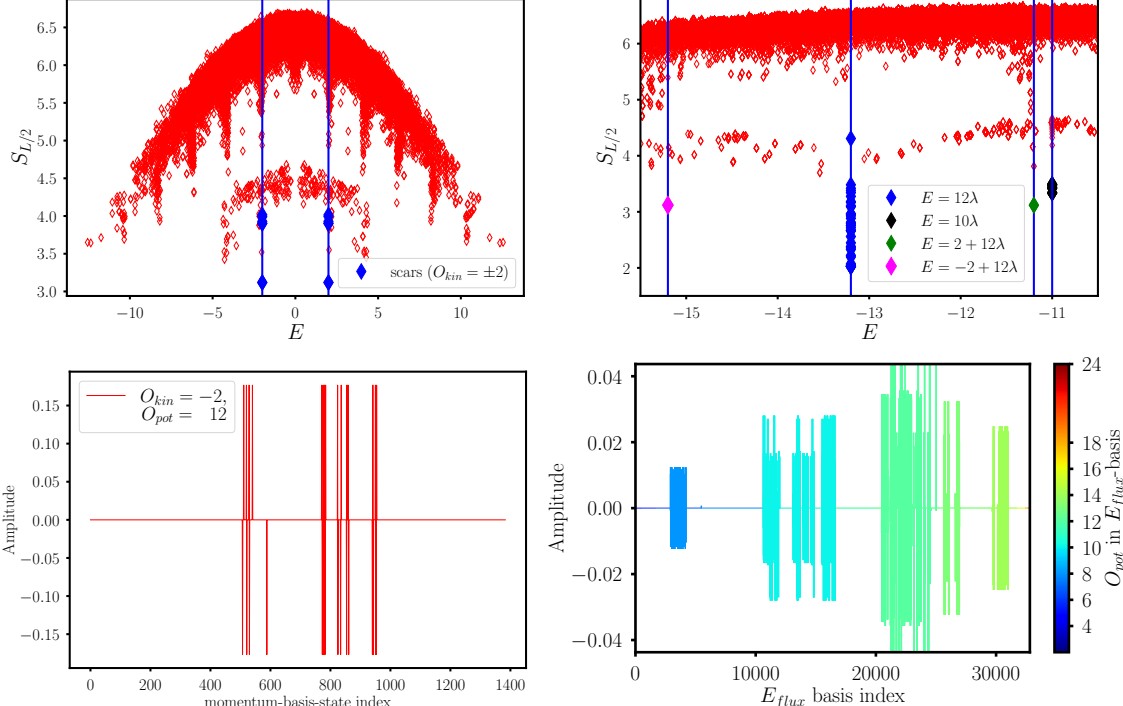

Figure 10: (top left) The bipartite entanglement entropy, $S_{L/2}$, shown for a system of dimension $(6,4)$ at zero winding number and $\lambda = 0$. The blue diamonds represent the eigenstates with $\mathcal{O}_{kin} = \pm 2$. (top right) $S_{L/2}$ for the same ladder but at $\lambda = -1.1$. Type-I, type-II, and type-IIIA scars are marked in the plot. (bottom left) The amplitudes shown as a function of the basis states for a type-IIIA scar at momentum $(0, 0)$. (bottom right) The amplitudes of a type-IIIB scar shown in the basis of the electric flux Fock states. The different colors represent the different $\mathcal{O}_{pot}$ values of the contributing Fock states.

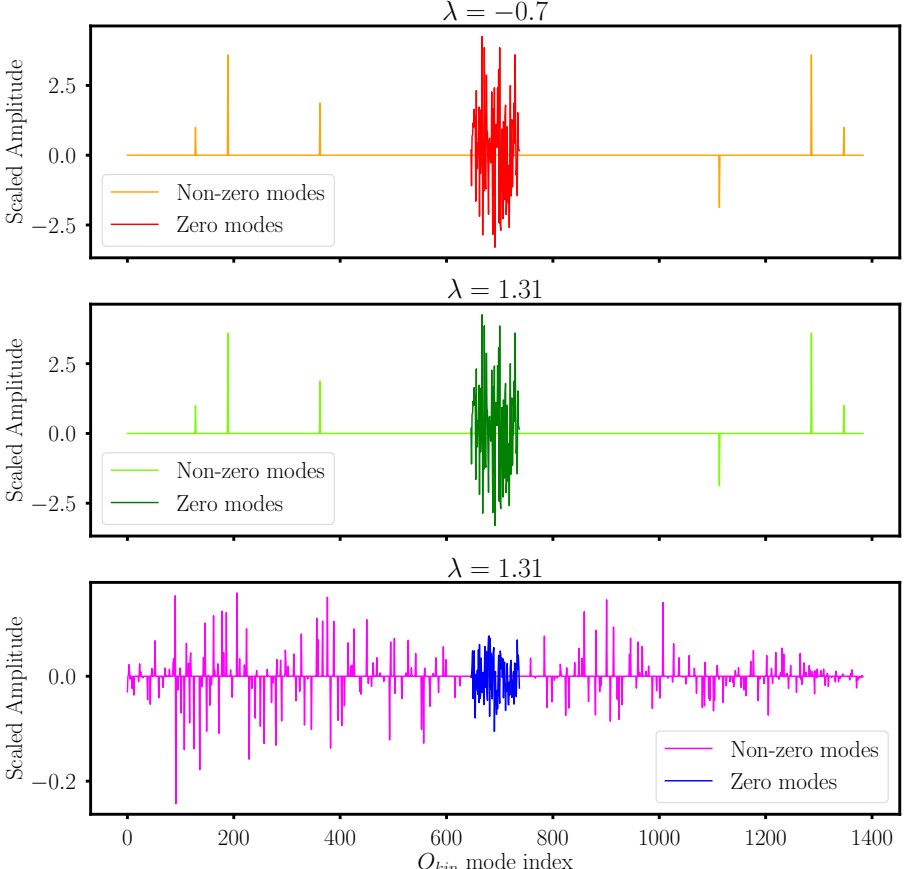

Figure 11: The top and the middle panels express a type-II scar with quantum numbers $(k_x, k_y, C) = (0, 0, -1)$ for the $(6, 4)$ system at zero winding number at $\lambda = -0.7$ and $\lambda = 1.31$ respectively in terms of the eigenmodes of $\mathcal{O}_{\text{kin}}$ obtained using ED. The bottom panel shows a nearby eigenstate with the same quantum numbers at $\lambda = 1.31$. The amplitudes of the zero modes and the nonzero modes of $\mathcal{O}_{\text{kin}}$ are rescaled separately such that the amplitude of a particular zero (nonzero) mode stays unchanged with $\lambda$.

**Order-by-disorder in the Hilbert space**: Lastly, we would like to stress a point already discussed at some length in Ref. [42]. As can be seen from Fig. 10 (top left panel), there are no outliers in $S_{L/2}$ at $E = 0$ for $\lambda = 0$ even though there are 46 (8) type-I scars with the corresponding $\mathcal{O}_{\text{pot}}$ eigenvalue being 12 (10) (Table 3). This is because the type-I scars, being anomalous zero modes, are degenerate with the other exponentially many non-anomalous zero modes at $\lambda = 0$. An arbitrary superposition of such type-I scars with these other zero modes is also an eigenstate that explains the absence of outliers in the numerical plot shown in Fig. 10 (top left panel). Turning on a finite $\lambda$ (Fig. 10 (top right panel)) makes these type-I scars appear an outliers of $S_{L/2}$ at energies $E = 12\lambda$ and $E = 10\lambda$ respectively by stabilizing these anomalous zero modes as new eigenstates at $\lambda \neq 0$ [42] while the other zero modes hybridize with the nonzero modes. These anomalous zero modes appear as pseudo-random superpositions in the basis of the zero modes of $\mathcal{O}_{\text{kin}}$ obtained from ED. However, these superpositions are completely different from an arbitrary superposition in the zero mode basis which is expected to satisfy the ETH and be delocalized in the Hilbert space. For example, Fig. 12 shows the amplitudes of two such quantum scars (one with $\mathcal{O}_{\text{pot}} = 6$ and another with $\mathcal{O}_{\text{pot}} = 8$) in the basis of the numerically obtained zero modes at $\lambda = 0$ for the system $(4, 4)$

whereby both scars appear as two different realizations of a pseudo-random superposition of such modes. However, the majority of such linear superpositions of the zero modes will not diagonalize $\mathcal{O}_{\text{pot}}$ and the type-I scars thus represent very special pseudo-random superpositions that are automatically highly localized in the Hilbert space. Thus, the appearance of the type-I scars at $\lambda \neq 0$ is akin to an "order-by-disorder" mechanism, but in the Hilbert space.

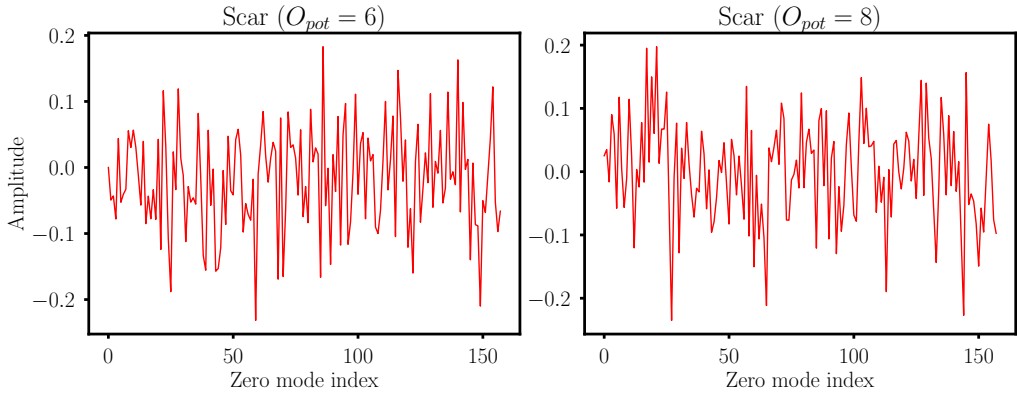

Figure 12: Two type-I scars, with $\mathcal{O}_{\text{pot}} = 6$ (8) in the left (right) panel, expressed in the basis of the zero modes of $\mathcal{O}_{\text{kin}}$ obtained from ED. Both the anomalous states appear as pseudo-random superpositions of the numerically obtained zero modes at $\lambda = 0$.

## 7.2 Results for the QDM

While there is still an exponentially large number of zero modes in system size at $\lambda = 0$, interestingly there are no type-I scars (or, the other varieties that we have discussed here) on $(L_x, 2)$ systems for the QDM based on numerical evidence. However, the behavior of the bipartite entanglement entropy, $S_{L/2}$, for all the eigenstates in the zero winding number sector $(W_x, W_y) = (0, 0)$ suggests that there are a set of eigenstates with low $S_{L/2}$ present at $\lambda = 0$, when one focuses in the range of $E/L \in [-0.2, 0.2]$ and $S_{L/2} \sim 2$, whose entanglement does not seem to follow a volume-law scaling (Fig. 13 (left panel)). These states vanish when $\lambda \sim O(1)$ as shown in Fig. 13 (right panel). Similarly, an analysis of the numerical data for the QDM in the systems with dimensions $(6, 6)$ and $(8, 6)$ (with the latter case being constrained to momenta $(k_x, k_y) = (0, 0), (0, \pi), (\pi, 0), (\pi, \pi), (\pi/2, 0), (0, \pi/2), (\pi/2, \pi/2)$) does not yield any scar solely made of the zero modes of $\mathcal{O}_{\text{kin}}$, nor does the $\mathcal{O}_{\text{kin}}$ operator have any nonzero integer or simple irrational numbers as eigenvalues for these system dimensions. However, as we will discuss below, anomalous zero modes and other scar varieties (type-IIIC scars) are present in the QDM with dimension $(L_x, 4)$.

**Type-I & type-IIIC scars**: For the $(4, 4)$ system, the QDM has 9 type-I scars with $\mathcal{O}_{\text{pot}} = 4$ and 1 type-I scar with $\mathcal{O}_{\text{pot}} = 6$. For the $(6, 4)$ system, the QDM has 6 type-I scars, each with $\mathcal{O}_{\text{pot}} = 4$. For an $(8, 4)$ system, the situation is even more interesting as summarized in Table 3. At $\lambda = 0$, there are 16 eigenvalues of $\mathcal{O}_{\text{kin}}$ of $\sqrt{2}$ $(-\sqrt{2})$ which are expected to be anomalous from the previous discussion in Section 5.1. Indeed, their bipartite entanglement entropy is much smaller than the neighboring eigenstates which shows the anomalous nature of these type-IIIC scars (Fig. 14, top left panel). These scars do not have a well-defined eigenvalue for $\mathcal{O}_{\text{pot}}$ and are, thus, eigenstates of $\mathcal{H}$ only when $\lambda = 0$. The zero modes are also indicated in the same figure which shows that these have high $S_{L/2}$ at $\lambda = 0$ due to the hybridization of the type-I scars with the other non-anomalous zero modes at this coupling. At $\lambda \neq 0$, the

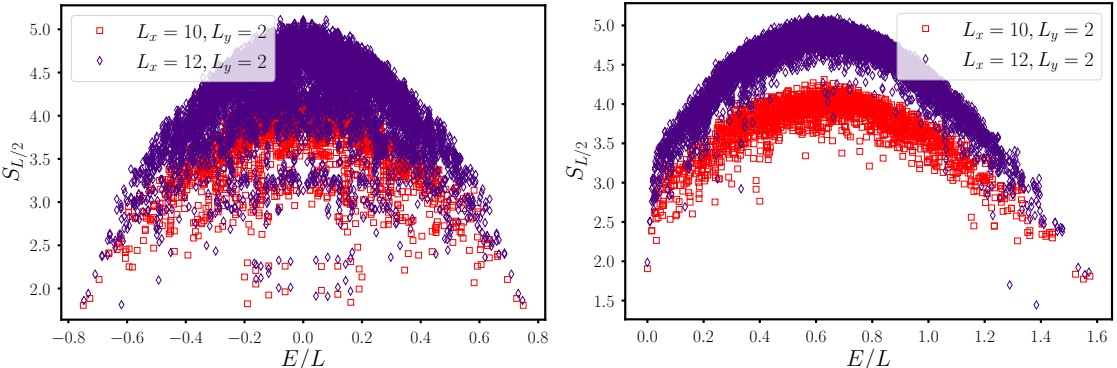

Figure 13: Bipartite entanglement entropy, $S_{L/2}$, shown for all the energy eigenstates for the QDM with dimensions $(10, 2)$ and $(12, 2)$ in the zero winding number sector. The left panel shows data for $\lambda = 0$ while the right panel shows data for $\lambda = 1$.

type-I scars show up as outliers of $S_{L/2}$ with there being 4 scars with $(\mathcal{O}_{\text{kin}}, \mathcal{O}_{\text{pot}}) = (0, 8)$, 16 scars with $(\mathcal{O}_{\text{kin}}, \mathcal{O}_{\text{pot}}) = (0, 7)$ and 8 scars with $(\mathcal{O}_{\text{kin}}, \mathcal{O}_{\text{pot}}) = (0, 4)$ (Fig. 14, top right panel). Expectation values of $S_1$ and $E_{corr}$ for the scar states (Fig. 14 bottom left and right) also appear as outliers from the ETH band. No other type-II or type-III scars were found at $\lambda \neq 0$. Using the method advocated in [87], it is possible to order the scar eigenstates in order of their increasing entanglement entropy. After ordering them, we obtain several eigenstates for which entanglement entropies are simple rational multiples of $\ln(2)$. This is an indication of a possibly simple description of these anomalous eigenstates and the analytic structure of some of these states will be discussed in the next Section.

The amplitudes of the type-I and the type-IIIC scars with the lowest entanglement entropy are shown in Fig. 15 in the basis of the electric flux Fock states. The corresponding amplitudes for a nearby eigenstate is also shown in each of these cases. Comparing these amplitude profiles, it is clear that the type-I and the type-IIIC scars are far more localized in the Hilbert space than the nearby eigenstates for the QDM. The type-I scars, in particular, have very few contributing electric flux Fock states, which again points towards a possible analytic description.

# 8 Lego scars in QDM: exact results

An obvious question that arises from the results so far is whether it is possible to analytically construct either of the different types of scars that we observe in the QLM or the QDM. While we do not know an analytic approach to construct all types of observed scars here, the examination of certain type-I scars for the QDM on systems with $L_y = 4$ expressed in the electric flux basis (after an entanglement entropy minimization algorithm implementation) gives an idea of how to construct such type-I scars analytically so that these are exact eigenstates of $\mathcal{H}$ at any $\lambda$. As we show, these states can be viewed most naturally in terms of basic building blocks, or legos, which have a dead zone at boundaries parallel to the $y$ direction and contain entangled units like emergent singlets and other more complicated structures in the interior. These legos can then be fitted with each other in the $x$ direction to create exact eigenstates in systems with arbitrarily large $L_x$. The numerically obtained type-I scars in the QDM (as summarized in Table. 3), barring a single scar with $\mathcal{O}_{\text{pot}} = 4$ and another with $\mathcal{O}_{\text{pot}} = 6$ for the $(4, 4)$ system, all turn out to be examples of such lego scars.

To see how this works out, consider a $S_{L/2}$ minimized type-I scar for the system $(L_{\text{x}}, L_{\text{y}})$ $= (6, 4)$, with $(\mathcal{O}_{\text{kin}}, \mathcal{O}_{\text{pot}}) = (0, 4)$. On expressing the scar wavefunction in the electric flux

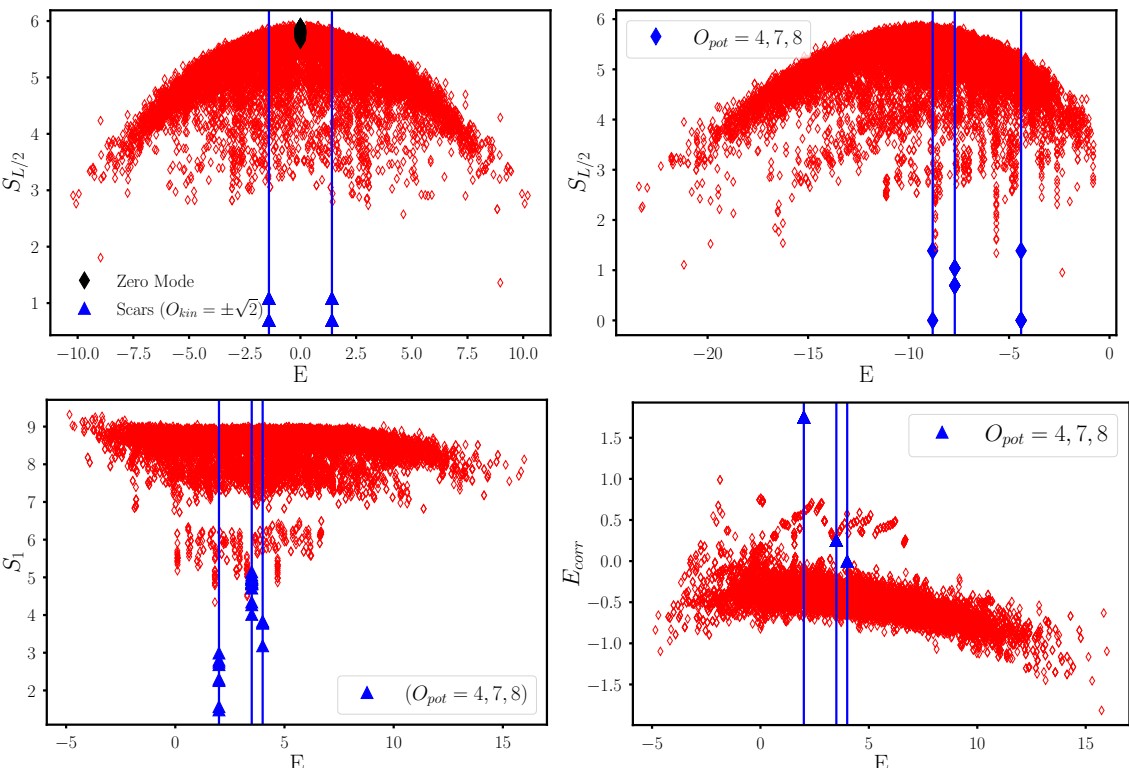

Figure 14: (Top) The bipartite entanglement entropy, $S_{L/2}$, shown for all the energy eigenstates at zero winding number for the $(8,4)$ QDM. The left (right) panel shows the data for $\lambda = 0$ ($\lambda = -1.1$). (Bottom panel) The Shannon entropy, $S_1$ and electric flux correlator $E_{\text{corr}}$ for the same lattice $(8,4)$ at a different $\lambda = 0.5$. The degenerate scars are not sorted using the EE minimization routine here. In all the figures, the vertical blue lines mark the value of the energy eigenvalue of the corresponding scar state.

basis, we see that only four basis states contribute with the same amplitude but with different signs, as illustrated in Fig. 16. It is clear that the basis states, $|c_1\rangle, \cdots, |c_4\rangle$, each have four flippable plaquettes (denoted in blue) while the rest of plaquettes are unflippable. Moreover, the horizontal bipartition (along axis $A_H$) yield $S_{L/2} = 0$, while the vertical bipartition (along the axes $A_V$) yield $S_{L/2} = 2\ln 2$. This suggests that the wavefunction can be expressed as a tensor product of two unentangled pieces along the vertical $L_x$ direction, which we call *legos*.

On closer examination, it is easy to identify the two legos $|\mathcal{L}_1\rangle, |\mathcal{L}_2\rangle$ drawn separately in Fig. 17. Each lego is three-plaquette rows thick, and the middle row is an equal *antisymmetric* superposition of a clockwise and an anti-clockwise plaquette separated by non-flippable plaquettes. This forms the active region of the lego, an emergent *singlet,* which under the action of $\mathcal{O}_{\text{kin}}$ produces different states that exactly cancel each other. The first and the third rows are inert, comprising of non-flippable plaquettes (which get trivially annihilated by $\mathcal{H}$). This inactive zone ensures that the excitations generated by $\mathcal{O}_{\text{kin}}$ while acting on the active zone cannot reach the boundary, and can only interfere among themselves. It is important to stress here that the inactive zone only exists because of the sign structure of the contributing states within the active zones. For example, if $\mathcal{H}$ is applied repeatedly to any one of the four contributing states in Fig. 16, then all other states in the same symmetry sector of the Hilbert space will be generated. The particular sign structure of the legos ensures that certain excitations destructively interfere before they can propagate out of an active zone. The reader is invited to check this for $|\mathcal{L}_1\rangle$, where $\mathcal{O}_{\text{kin}}$ acts non-trivially on the flippable plaquettes, and gives rise to four

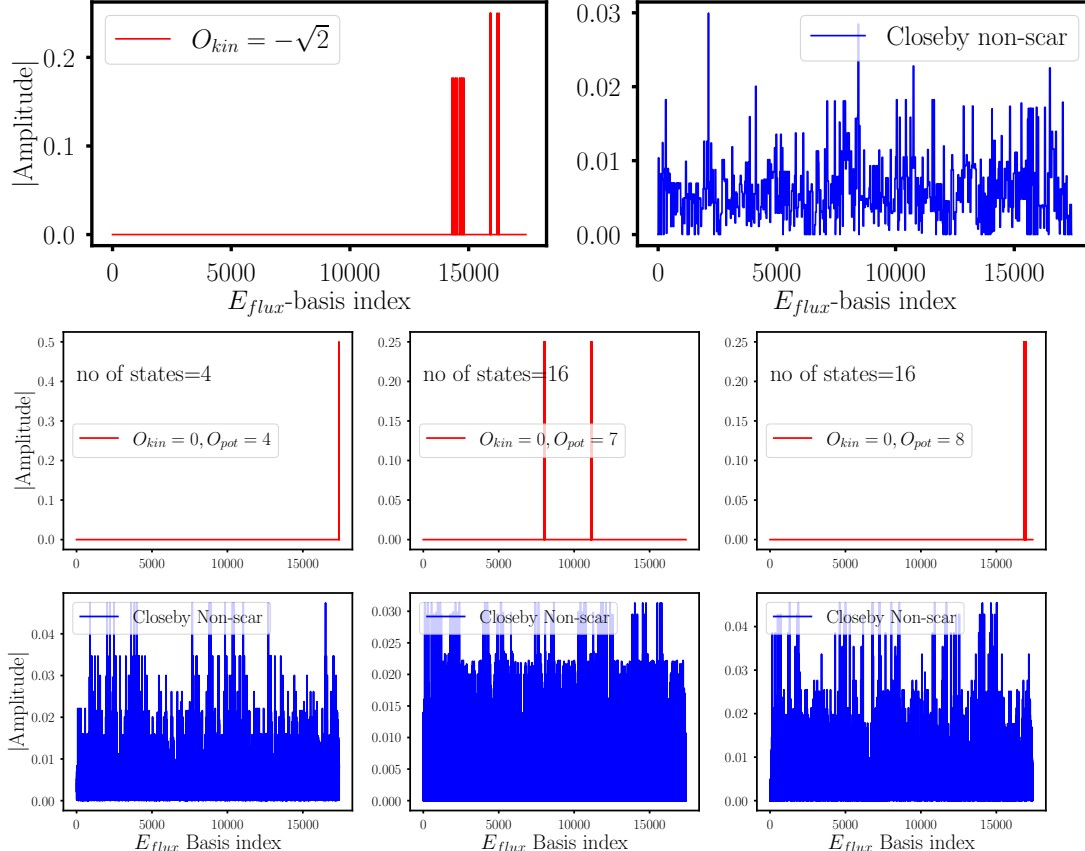

Figure 15: (Top panel) The amplitudes of a type-IIIC scar (a neighboring eigenstate) shown in the basis of the electric flux Fock states for a system of dimension $(8,4)$ at zero winding number in the left (right) plot. (Middle panel) The amplitudes of the type-I scar with the lowest entanglement entropy with $\mathcal{O}_{\text{kin}} = 4$ (left plot), $\mathcal{O}_{\text{pot}} = 7$ (middle plot) and $\mathcal{O}_{\text{pot}} = 8$ (right plot) shown in the basis of the electric flux Fock states for the same lattice dimension. The number of contributing electric flux Fock states is also indicated inside each of these three figures. (Bottom panel) The amplitudes of a neighboring eigenstate corresponding to each of the three type-I scars in the middle panel shown in the three figures.

states which cancel each other. Two of these states, with four flippable plaquettes with alternating circulations along a line in the active zone, would have taken the excitations through the inactive zones if they did not cancel each other due to the sign structure adopted in $\mathcal{L}_1$. Another way of stating this is to note that in Fig. 17, the inert zones do not contain any flippable plaquettes iff the active zone contains a single clockwise and and a single anticlockwise plaquette separated by non-flippable plaquettes as shown. A configuration with alternating clockwise and anticlockwise plaquettes in the active zone with no separating plaquettes in between would create flippable plaquettes in the inactive zone as well. Moreover, the dashed lines indicate the missing links in the lego $|\mathcal{L}_1\rangle$ which fit vertically with the lego $|\mathcal{L}_2\rangle$. Thus, the scar state of Fig. 16 can be expressed as $|\psi_s\rangle = |\mathcal{L}_1\rangle \otimes |\mathcal{L}_2\rangle$, with the normalization taken into account as explained in Fig. 17. Moreover, this analytic construction immediately shows that one should expect 6 such scars with $(\mathcal{O}_{\text{kin}}, \mathcal{O}_{\text{pot}}) = (0,4)$, corresponding to the different arrangements of the singlets, and which matches with the data in Table 3. We can further predict the $S_{L/2}$ for the different cases: whenever the axis cuts the (two) singlets one gets $2\ln 2$, while all other cases yield 0.

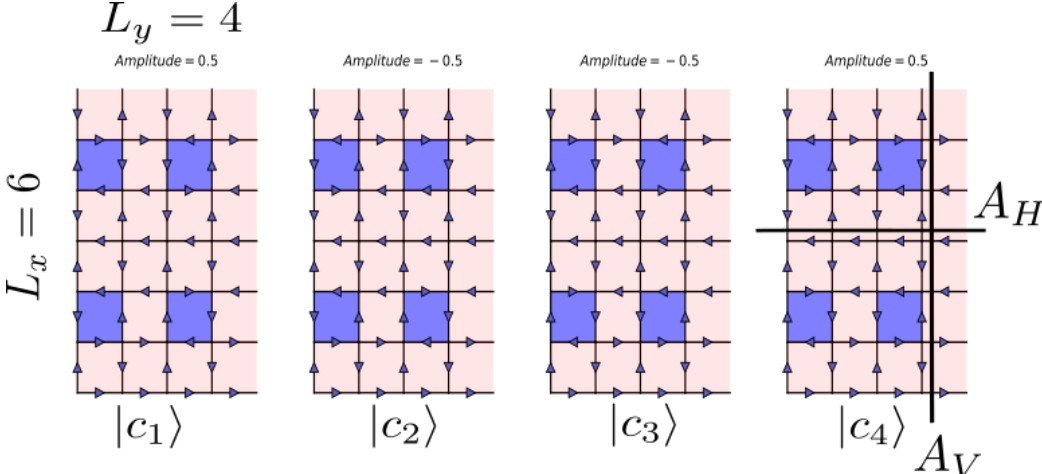

Figure 16: A type-I $S_{L/2}$ minimized scar for the QDM on $L_x = 6, L_y = 4$ and with $(\mathcal{O}_{\text{kin}}, \mathcal{O}_{\text{pot}}) = (0, 4)$. It is a linear superposition of four basis states $|c_1\rangle$, $|c_2\rangle$, $|c_3\rangle$, and $|c_4\rangle$ with equal amplitudes but differing signs, each with four flippable plaquettes which are shaded in blue. The $S_{L/2}$ of the scar is 0 for a horizontal bipartition (axis $A_H$), and $2\ln 2$ for a vertical bipartition (axis $A_V$). Only one bipartition axis is shown in each case. Due to PBC, there is another identical axis exactly halfway across the linear extent of the lattice in each direction.

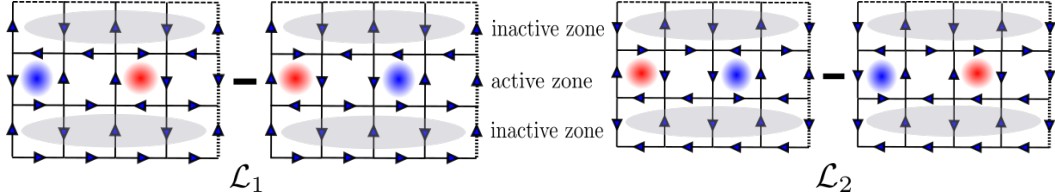

Figure 17: Legos $|\mathcal{L}_1\rangle$, $|\mathcal{L}_2\rangle$ which form all the type-I scars of the $(L_x, L_y) = (6, 4)$ QDM. The active zone is a quantum superposition of a clockwise (red) and an anti-clockwise (blue) plaquette, which we refer to as a *singlet*. The active zone is shielded by a row of inert unflippable plaquettes (grey shaded area), which trivially vanishes under the action of $\mathcal{H}$. The action of $\mathcal{O}_{\text{kin}}$ on the active zone causes a quantum superposition of different flux states which cancel each other. Simultaneously, the arrangement of links in the inactive zone together with the sign structure of the active zone ensures that no excitation is able to propagate in the vertical direction, effectively caging the excitations. We note that when this lego is inserted in a wavefunction it has to be appropriately normalized. Each singlet lego contributes a factor of $\frac{1}{\sqrt{2}}$, and the normalization in the wavefunction due to $n$ singlets is $(\frac{1}{\sqrt{2}})^n$. Furthermore, the lego $|\mathcal{L}_1\rangle$ will not fit on itself (and similarly for $|\mathcal{L}_2\rangle$), it needs to be alternated with a different compatible lego.

This construction also works out for the type-I scars $(\mathcal{O}_{\text{kin}}, \mathcal{O}_{\text{pot}}) = (0, 4)$ on the larger lattice $(L_x, L_y) = (8, 4)$. Once again, the contribution comes from four basis states, which can be constructed using two legos $|\mathcal{L}_3\rangle$, $|\mathcal{L}_4\rangle$ as illustrated in Fig. 19. As in the previous case, there are two parts: (i) the active part (the same *singlet*), which transforms nontrivially under the $\mathcal{O}_{\text{kin}}$ but vanishes under quantum superposition, exactly as before, (ii) an inactive part, which trivially vanishes under $\mathcal{O}_{\text{kin}}$. Piecing together these legos such that the active parts are shielded by the dead parts *cages* the excitations caused by $\mathcal{O}_{\text{kin}}$. This phenomenon is thus an example of *quantum caging*.

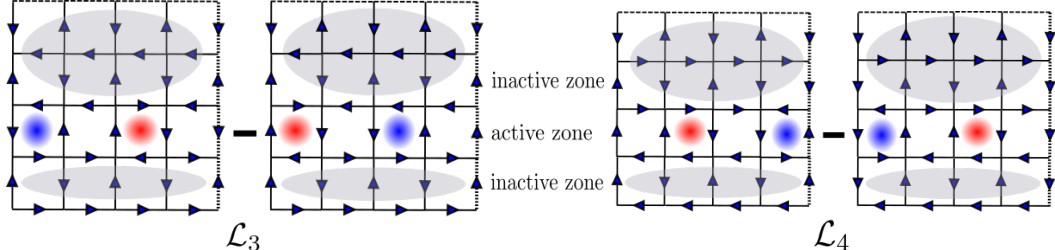

Figure 18: The legos which form the type-I scars $(\mathcal{O}_{\text{kin}}, \mathcal{O}_{\text{pot}}) = (0, 4)$ of the system $(L_{\text{x}}, L_{\text{y}}) = (8, 4)$. The structure of the legos $|\mathcal{L}_3\rangle, |\mathcal{L}_4\rangle$ is the similar to that of $|\mathcal{L}_1\rangle, |\mathcal{L}_2\rangle$. The inactive part is thicker in this case, and serves the same purpose as explained before. Moreover, the normalization factors and the fitting of the respective legos occur just as for $|\mathcal{L}_1\rangle, |\mathcal{L}_2\rangle$.

It turns out that the lattice $(L_{\text{x}}, L_{\text{y}}) = (8, 4)$ also has other type-I scars with more exotic legos, that can be analytically understood. For example, the 16 scars with $(\mathcal{O}_{\text{kin}}, \mathcal{O}_{\text{pot}}) = (0, 7)$ can again be realized with a lego wavefunction, consisting of a fatter structure (that cannot be decomposed in terms of singlets) embedded within two singlets on either side and separated by an one row thick inactive zone. As usual, these singlets are themselves separated from the boundary with one row of inactive plaquettes. This lego is illustrated in Fig. 19, with the respective normalization of the constituent elements. Acting $\mathcal{O}_{\text{kin}}$ on the fatter structure leads to twelve states that exactly cancel among each other in a pairwise manner due to the sign profile shown in Fig. 19. Once again, the degeneracy of scars produced using such legos can be counted by the different positions of the flippable plaquettes in the active zone. Intriguingly,

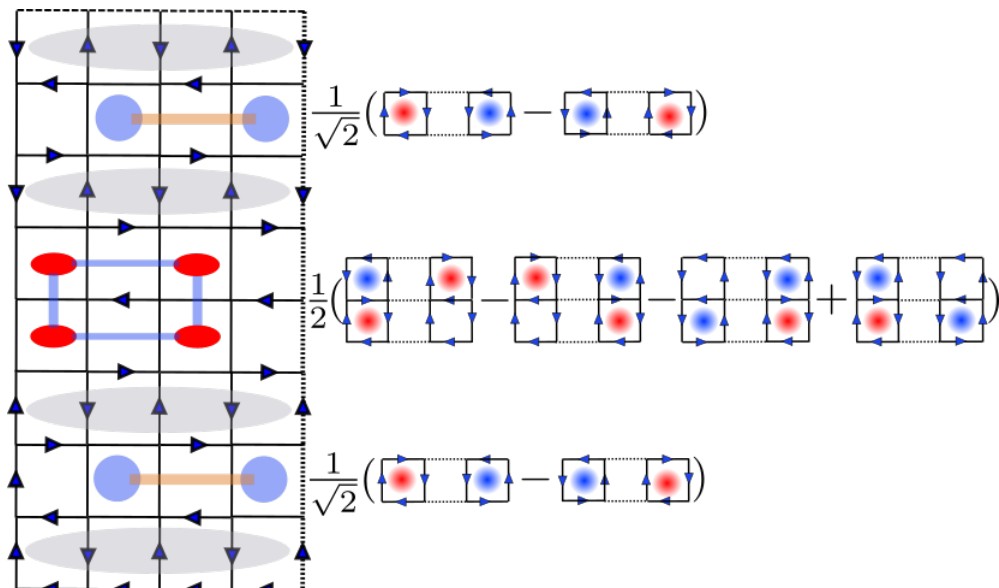

Figure 19: The lego forming the scars of the lattice $(L_{\text{x}}, L_{\text{y}}) = (8, 4)$ with $(\mathcal{O}_{\text{kin}}, \mathcal{O}_{\text{pot}}) = (0, 7)$. This comprises of three active zones, two of which are the singlets introduced earlier. The third is a two row thick structure comprising of three flippable plaquettes, resonating on four different plaquettes, and expressed as a quantum superposition with four electric flux states (with a normalization factor $\frac{1}{2}$). The singlets are separated from the boundary with a single layer thick inactive zone, which prevents the excitations generated by $\mathcal{O}_{\text{kin}}$ from vertical propagation.



the entanglement entropy $S_{L/2}$ (along a horizontal or a vertical bipartition) gets a contribution of an integer multiple of $\ln(2)$ or half-integer multiple of $\ln(2)$ depending on whether it cuts the singlet or the fatter structure, and on whether the flippable plaquette is touched.

It is also possible to obtain a close-packing of singlets which can explain the remaining type-I scar on the lattice $(L_x, L_y) = (8, 4)$ with $(\mathcal{O}_{kin}, \mathcal{O}_{pot}) = (0, 8)$. This lego, shown in Fig. 20 consists of two singlets, separated by a row of inactive plaquettes and horizontally displaced by one lattice spacing. Finally, as in all the above cases, there is the one row thick of inactive zone separating the singlets from the boundary. Naturally, these structures are realized even on smaller lattices $(L_x, L_y) = (4, 4)$ which was also verified numerically. Since $x$ and $y$ directions are equivalent for the system $(L_x, L_y) = (4, 4)$, the singlets can be arranged together either horizontally or vertically which gives a degeneracy of 8.

All the discussion above shows that it is possible to construct an exponentially large number of eigenfunctions for the QDM on arbitrarily large lattices as long as one dimension is fixed to 4, and the other is a multiple of 4 or 6. The generic form of the eigenfunction is $|\psi_s\rangle = |\mathcal{L}_i\rangle \otimes |\mathcal{L}_j\rangle \otimes \cdots |\mathcal{L}_k\rangle$, where the legos are of the type described above, and we just need to ensure to choose the type of lego that can fit with each other at the boundaries to create type-I scars. For example, 4 of the type-I scars for the QDM in a system with dimensions $(L_x, L_y) = (8, 4)$ are created by patching two identical legos where the lego type is shown in Fig. 20. Such lego scars satisfy area law by construction even in the thermodynamic limit and thus necessarily violate the ETH. This is a remarkable first analytic result for anomalous zero modes for this very well-known model, which is also a strongly interacting theory. Moreover, it is curious that these types of scars give rise to excitations that exhibit sub-dimensional motion, reminiscent of fractonic models [88]. Whether more complicated legos with richer internal structures and for $L_y > 4$ exist is left as an open problem for a future study.

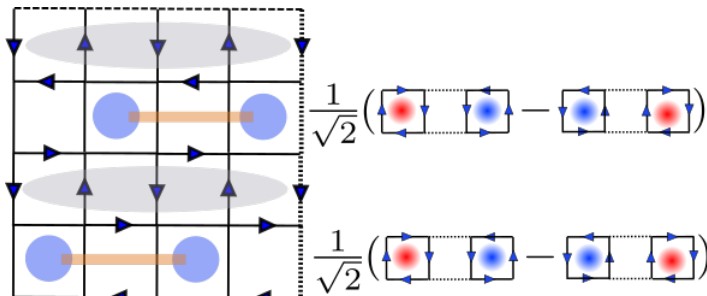

Figure 20: The lego which has singlets placed on alternate rows, each separated by an inactive zone to allow excitations to propagate only in the horizontal direction.

# 9 Scars in non-zero winding sectors of QDMs and QLMs

From Fig. 5, it is clear that an exponentially large number of zero modes are also present for nonzero winding number sectors in the QLM as well as in the QDM. A natural question is whether anomalous zero modes that are simultaneous eigenkets of both $\mathcal{O}_{kin}$ and $\mathcal{O}_{pot}$ exist for the other topological sectors characterized by nonzero $W_x, W_y$.

This is indeed the case as our numerical results show. In Fig. 21, we show evidence for a type-I scar with $(\mathcal{O}_{kin}, \mathcal{O}_{pot}) = (0, 8)$ for the winding number sector $(W_x, W_y) = (1, 0)$ for a QLM defined on the $(6, 4)$ system. The left panel shows the bipartite entanglement entropy, $S_{L/2}$, which clearly shows that this state is an outlier in $S_{L/2}$ compared to the neighboring eigenstates at a coupling $\lambda = -1.1$. The right panel shows the amplitudes of this type-I scar in the basis of the electric flux Fock states from which it is evident that this state is anomalous.

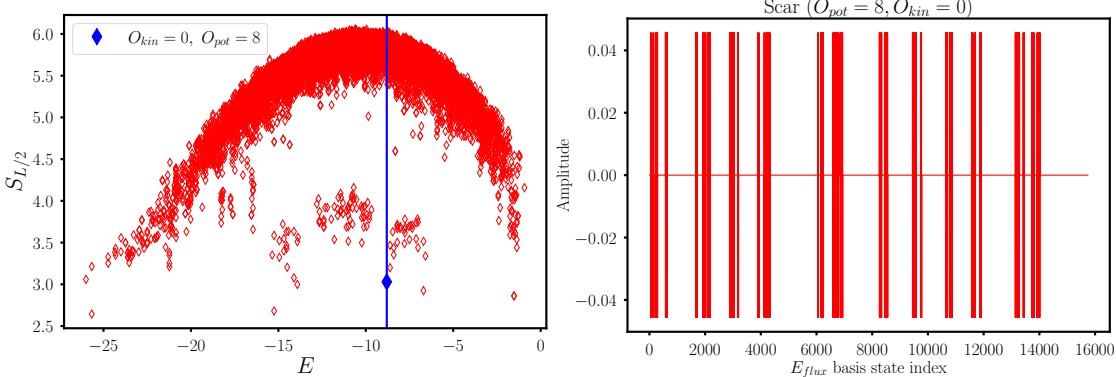

Figure 21: (Left) The bipartite entanglement entropy, $S_{L/2}$, for all the eigenstates of a QLM on the lattice $(6, 4)$ at $\lambda = -1.1$ in the winding number sector $(W_x, W_y) = (1, 0)$. A single type-I scar shows up as an outlier in entanglement entropy as shown. (Right) The amplitudes of the same type-I scar expressed in the basis of the electric flux Fock states.

In Fig. 22, we show another example of type-I scars at nonzero winding numbers for the QDM on $(8, 4)$ lattice at a coupling of $\lambda = -1.1$ for a winding of $(W_x, W_y) = (1, 0)$. In this example, there are 8 degenerate type-I scars with $(\mathcal{O}_{kin}, \mathcal{O}_{pot}) = (0, 4)$. The left panel shows the 8 scars before entropy entanglement minimization while the right panel shows the data after the minimization that leads to 4 type-I scars with $S_{L/2} = 0$ and 4 others with $S_{L/2} = \ln(2)$.

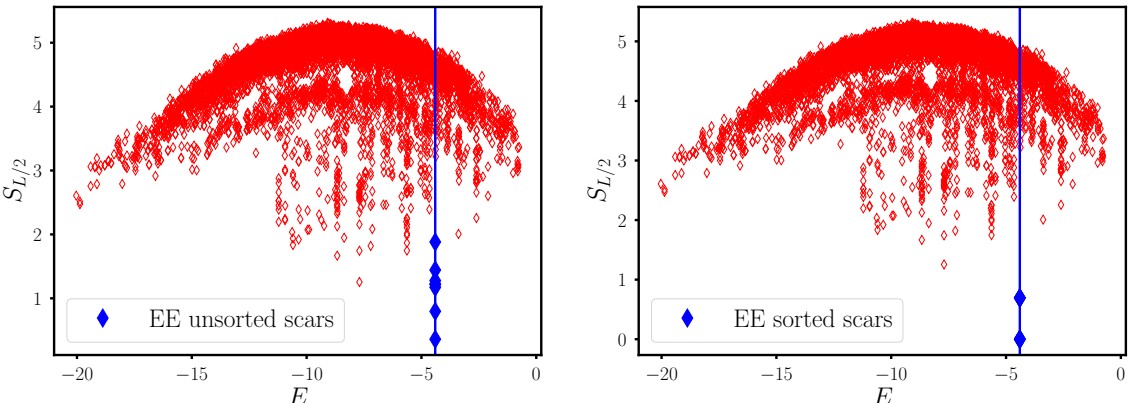

Figure 22: Bipartite entanglement entropy $S_{L/2}$ for each eigenstate of the QDM in the $(W_x, W_y) = (1, 0)$ winding number sector with $\lambda = -1.1$ for a system with dimensions $L_x = 8$, $L_y = 4$. The blue dots in both panels represent the 8 quantum scars that have a well-defined $(\mathcal{O}_{kin}, \mathcal{O}_{pot}) = (0, 4)$. The right panel shows $S_{L/2}$ after implementing an entanglement entropy minimization procedure to sort the scars in terms of increasing entanglement entropy since any linear combination of these degenerate scars is also an eigenstate.

## 10 Conclusions and outlook

We have shown the existence of a rich variety of quantum many-body scars in the well-known $U(1)$ quantum link model and the quantum dimer model with a Rokhsar-Kivelson Hamilto-

nian $\mathcal{H} = \mathcal{O}_{\mathrm{kin}} + \lambda \mathcal{O}_{\mathrm{pot}}$ defined using the elementary square plaquettes on finite square and rectangular systems with periodic boundary conditions. Such models also represent non-trivial Abelian lattice gauge theories without dynamical matter fields. A particular limit of both these models ($\lambda = 0$), when the Hamiltonian only contains off-diagonal terms in the computational (electric flux) basis, supports exact mid-spectrum zero modes whose number grow exponentially with system size due to an index theorem. The spectrum can thus be decomposed into zero and nonzero modes in this strongly interacting limit. The quantum many-body scars have a natural classification in terms of these modes even when the diagonal terms in the Hamiltonian are made non-zero. All the possibilities, i.e., scars composed of solely the zero modes of $\mathcal{O}_{\mathrm{kin}}$, solely the nonzero modes of $\mathcal{O}_{\mathrm{kin}}$ and an admixture of both zero and nonzero modes of $\mathcal{O}_{\mathrm{kin}}$ are realized in these models. Some of these scars are eigenstates of $\mathcal{H}$ only for $\lambda = 0$ while the other varieties of scars are eigenstates for all $\lambda \neq 0$ and all $\lambda$, respectively. Some of these types of scars have been reported, albeit numerically on finite lattices, for the first time in a non-integrable model to the best of our knowledge.

Some special linear combinations of the zero modes of $\mathcal{O}_{\mathrm{kin}}$ also diagonalize $\mathcal{O}_{\mathrm{pot}}$ and turn out to be much more localized in the Hilbert space compared to the other zero modes in both these models. These anomalous zero modes survive as eigenstates even when $\lambda \neq 0$ while the other zero modes hybridize with the nonzero modes providing an example of "order-by-disorder", but in the Hilbert space. For the quantum dimer model on specific rectangular lattices with a fixed $L_{\mathrm{y}} = 4$ and $L_{\mathrm{x}}$ which is a multiple of 4 or 6, but otherwise arbitrary, we have constructed analytic examples of such anomalous zero modes that we dub as lego scars. Such states can be expressed as tensor products of the basic building blocks, or legos, that contain emergent singlets and other more complicated entangled units where these units can be fit to each other at the boundaries. Since there is more than one variety of such legos, there exists an exponentially large number of such lego scars when $L_{\mathrm{y}} = 4$ and $L_{\mathrm{x}} \gg 1$. We conjecture that the $U(1)$ quantum link model also hosts an exponential number of anomalous zero modes when $L_{\mathrm{x}} \gg 1$ and $L_{\mathrm{y}} = 4$ based on the numerical trend of there being a larger number of anomalous zero modes in the $U(1)$ quantum link model compared to the quantum dimer model in lattices of dimensions $(L_{\mathrm{x}}, 4)$.

Several open questions arise from our study. While the lego scars provide an analytically tractable example of anomalous zero modes in the quantum dimer model, it will be interesting to see whether some of the anomalous zero modes in the $U(1)$ quantum link model can also be expressed in a closed form. Similarly, an analytic understanding is required for the other varieties of scars that require nonzero modes of $\mathcal{O}_{\mathrm{kin}}$. Whether anomalous zero modes and other quantum scars arise in higher spin representations of the $U(1)$ quantum link model deserves a separate investigation. Finally, the index theorem present at $\lambda = 0$ in both the models can be broken by adding other non-commuting terms not considered in this study. For example, when $L_{\mathrm{y}} = 2$ for the quantum dimer model, the $\mathcal{O}_{\mathrm{pot}}$ interaction term fails to generate anomalous zero modes and this particular rectangular geometry with $L_{\mathrm{y}} = 2$ can be used to check whether other non-commuting terms may be added to $\mathcal{O}_{\mathrm{kin}}$ to generate order-by-disorder in the Hilbert space. It remains to be seen whether adding a specific class of further interactions to non-integrable models with an exponentially large manifold of mid-spectrum zero modes may provide a general route to quantum many-body scarring.

## Acknowledgements

We thank Krishnendu Sengupta for useful discussions. We would like to thank Emilie Huffman and Lukas Rammelmüller for agreeing to share the python scripts used to create Fig. 16.

# A  Entanglement minimization algorithm

Due to the singular value decomposition (SVD), a quantum state can be expressed through a tensor product of its constituent subsystems:

$$|\psi\rangle = \sum_{i=1}^{n} c_i^{\psi} |e_i\rangle = \sum_{i=1}^{n_A} \sum_{j=1}^{n_B} c_{i,j}^{\psi} |e_i^A\rangle \otimes |e_j^B\rangle = \sum_{i=1}^{n_S} \xi_i^{\psi} |\psi_A^i\rangle \otimes |\psi_B^i\rangle \,, \tag{14}$$

where $|e_i\rangle$ is an electric-flux state of the total system, and $|e_i^{A(B)}\rangle$ is a flux-state belonging to the sub-system A(B). $n_A$ and $n_B$ are the total number of basis states in the two subsystems and $c_{i,j}^{\psi}$ are the elements of a rectangular matrix of dimensions $n_A \times n_B$. From the SVD one obtains the real and non-negative Schmidt values $\xi_i^{\psi}$ with $i = 1, \cdots, n_S$ and $n_S = \min(n_A, n_B)$. The reduced density matrix of the system is:

$$\rho_A^{\psi} = \mathrm{Tr}_B(|\psi\rangle \langle\psi|) = \sum_{i=1}^{n_S} (\xi_i^{\psi})^2 |\psi_A^i\rangle \langle\psi_A^i| \,. \tag{15}$$

The corresponding $\alpha$-Renyi Entropy can be expressed as:

$$S_\alpha(\rho^{\psi}) = \frac{\log(\mathrm{Tr}[(\rho^{\psi})^{\alpha}])}{1-\alpha} = \frac{2\alpha}{1-\alpha} \log\left[ \left( \sum_{i=1}^{n_S} (\xi_i^{\psi})^{2\alpha} \right)^{1/2\alpha} \right] = \frac{2\alpha}{1-\alpha} \log(||\psi||_{2\alpha, n_S}) \,. \tag{16}$$

In the last expression, $||\psi||_{2\alpha, n_S}$ refers to Schmidt norm, which is defined from the SVD of the concerned pure-state $|\psi\rangle$ as: $||\psi||_{p,k} = \left( \sum_{i=1}^{k} (\xi_i^{\psi})^p \right)^{1/p}$, $(p \geq 1, k \leq n_S)$. The von Neumann entropy is obtained the limit $\alpha \to 1$, and given as $\sum_{i=1}^{n_S} -(\xi_i^{\psi})^2 \log(\xi_i^{\psi})^2$.

To maximize the Schmidt norm over the manifold of vectors belonging to a subspace (which is equivalent to minimizing the von Neumann entropy in the same subspace) we have implemented the algorithm proposed in [87]. Even though the algorithm does not guarantee convergence to the global maximum, but convergence to a local maxima can be proven [87] with this algorithm. The steps of the algorithm are as follows:

- Choose a random state $|\psi\rangle$ belonging to the concerned subspace.

- Apply SVD to $|\psi\rangle$ as described in Eq. (14).

- Let $P$ be the projection operator into the concerned subspace. Define the modified vector:
$$|\phi\rangle := \frac{P\left( \sum_{i=1}^{k} \left(\xi_i^{\psi}\right)^{p-1} |\psi_A^i\rangle \otimes |\psi_B^i\rangle \right)}{||P\left( \sum_{i=1}^{k} \left(\xi_i^{\psi}\right)^{p-1} |\psi_A^i\rangle \otimes |\psi_B^i\rangle \right)||} \,.$$

- Replace $|\psi\rangle$ by $|\phi\rangle$, and repeat from step-2.

To ensure better approach to a local maxima closer to the global one, we started with four randomly chosen vectors belonging to the subspace, and let the algorithm run for each of them, and accepted the end result which has the highest Schmidt norm. Convergence during each execution was ensured by letting the iteration loop run until the standard deviation calculated over the outcomes of last six iterations becomes smaller than a chosen threshold. Further, to perform the minimization, we have found the sequence $\alpha_n = 1 + \frac{1}{2^n}$ to give a sufficiently good saturation of the entanglement entropy (EE) minimized using this algorithm. The saturation is again tracked using the standard deviation over the last six iterations, until an iteration dependent threshold ($\frac{0.1}{n}$) is reached.

We can however do better in the search for global minima, by determining an orthonormal basis of this subspace comprised of least entangled states. This can be done by projecting out the EE minimized state from the subspace, run the entire algorithm again over the new subspace, and keep repeating until a complete basis is obtained. An advantage of such EE ordering is that it can compensate, to some extent, for the uncertainty of the algorithm to hit the global minimum of EE. A caveat of this method though, is that the resultant state from $n$-th execution of the algorithm may not necessarily be the $n$-th least entangled state of the subspace, therefore a last rearrangement is required on all the outcomes.

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
