# Peer review of "Scars from protected zero modes and beyond in $U(1)$ quantum link and quantum dimer models"

_SciPost Physics, doi:SciPost Phys. 12, 148 (2022)_

## Round 2 · Referee Report · Anonymous (Referee 1) · 2022-3-8

Strengths

- Comprehensive introduction to quantum link and quantum dimer models
- state-of-the-art numerics
- analytical understanding of some quantum scars

Weaknesses

- Novelty and importance of results not entirely clear
- Presentation of many numerical results with only little physical interpretation
- In parts hard to follow the discussion of scars for multiple different parameter choices

Report

The manuscript by Biswas et al. studies the occurrence of quantum many-body
scars in quantum link and quantum dimer models with different rectangular
geometries. Using large-scale exact diagonalization, they present evidence for
anomalous eigenstates in different symmetry sectors of the models, and provide
a classification of these anomalous states into distinct categories. Moreover,
for a subset of these states, an analytical construction in presented.

The field of quantum many-body scars has been exceptionally active during the
last couple of years. Understanding the origin of such nonthermal eigenstates,
studying their detailed aspects, and exploring their occurrence in different
models is a timely and interesting subject. With their work, the authors
contribute to this ongoing effort by providing an in-depth analysis of scars in
the context of lattice gauge theories, which have gained recent popularity also
in the condensed-matter community. The numerics is of state-of-the art level,
the analysis is thorough and fully sound, and the language is clear. I believe
that the manuscript can be of interest to researchers working on various
aspects of ergodicity-breaking in quantum many-body systems, as well as
to researchers who are new to the field.

I support to proceed towards the publication of the manuscript. While the
manuscript clearly fulfills all acceptance criteria of Scipost Physics Core, I
am unsure however if the novelty of the results warrants publication in Scipost
Physics, not least as many of the findings presented here appear to have
already been anticipated in a previous work of the authors (as also stated in
the abstract). Moreover, the manuscript presents an abundance of numerical
results, but often it is hard to follow the different parameter choices (I
appreciate the authors' attempt to support understandability by providing
overview tables) and, more importantly, physical interpretation of the obtained
results is scarce. In my opinion, it should be emphasized more clearly
what are the new results that substantially go beyond their previous work and
their importance should be explained. Likewise, the manuscript and its readers
would benefit if the authors would center their narrative more concisely around
the main new finding (see also next paragraph).

Reading the manuscript, it feels to me that the most important new result is
the analytical construction in terms of the "lego scars" (I am happy to be
convinced by the authors if they disagree with this). However, the
way the paper is written right now, this section almost gets lost since it is
far to the end of the paper, after an extensive discussion of numerical findings
of different types of quantum scars whose significance is less clear and
unexplained. For instance, one important aspect in the study of quantum scars
that comes to mind is dynamics. Can the authors comment on whether it is
possible to find simple physical initial states that show untypical dynamics
due to the scars? Is there a relation between the scars that the authors
observe and studies of disorder-free many-body localization in such lattice
gauge theories? Overall, in my opinion, the authors should strongly consider to
restructure the paper and focus more on the analytical construction, and maybe
think about delegating some less important aspects to the appendix.

Apart from this issue, I also have a number of comments and questions that I
would like the authors to take into account in their revision:

- In the introduction, the authors mention that constrained Hilbert spaces
appear to play an important role in the study of quantum scars. While QLM and
QDM both have constrained Hilbert spaces, the constraint is even stronger in
the QDM. Yet, it appears that the authors find fewer scars in this case. Do the
authors have a physical intuition for this finding?

- Can the authors explain their choice of \lambda for the presented numerical
results? Why is the level spacing in Fig. 6 shown for \lambda = 0.1 and 3.13,
but results later are shown for other values?

- Regarding the level spacing in Fig. 6, the apparent deviation from GOE
statistics in the QDM for larger \lambda should be explained more clearly.
I assume for \lambda \to \infty, one expects Poisson statistics?! At the
same time, the authors state that the model is nonintegrable for
\lambda ~ 1. However, in Fig. 6 (right, bottom) it seems like increasing the
systems size causes the statistics to drift towards Poisson?! This can be
important also for the analysis of the quantum scars later in the paper. If the
level statistics does not follow GOE behavior, the occurrence of anomalous
eigenstates might be less surprising.

- Would the analytical "lego-scar" construction also be possible for multiples
L_y = 4 \times n?

- On page 7, the authors write that they denote both models by {\cal H}_RK. On
page 9, they write that they refer to both models by {\cal H}.

- Are the lattices in Fig. 4 to be interpreted with PBC, i.e., are top and
bottom row (left and right column) equivalent?

- I appreciate the authors efforts to provide a sketch of the different types
of scars in Fig. 7. I must admit however, that I don't fully understand all
aspects, for instance does the number of arrows has a meaning?

Requested changes

See report.

  • validity: top
  • significance: good
  • originality: good
  • clarity: high
  • formatting: good
  • grammar: perfect

Author:  Arnab Sen  on 2022-03-29  [id 2334]

(in reply to Report 1 on 2022-03-08)

Response to Referee 1
==================================================================
We thank the Referee for taking the time and effort to carefully review our
manuscript. We are happy that they find the introduction comprehensive, the
numerics advanced, and the analytic understanding of some scars noteworthy.
We also appreciate pointing out the weak points of the current paper, and
we have improved and streamlined the presentation considerably to meet the
standards of SciPost physics. Please find our responses to the specific points
that were raised. Please also see the List of Changes to the manuscript,
implemented to address the comments.

>>I support to proceed towards the publication of the manuscript. While the
>>manuscript clearly fulfills all acceptance criteria of Scipost Physics Core,
>>I am unsure however if the novelty of the results warrants publication in
>>Scipost Physics, not least as many of the findings presented here appear to
>>have already been anticipated in a previous work of the authors (as also
>>stated in the abstract). Moreover, the manuscript presents an abundance of
>>numerical results, but often it is hard to follow the different parameter
>>choices (I appreciate the authors' attempt to support understandability by
>>providing overview tables) and, more importantly, physical interpretation of
>>the obtained results is scarce. In my opinion, it should be emphasized more
>>clearly what are the new results that substantially go beyond their previous
>>work and their importance should be explained. Likewise, the manuscript and
>>its readers would benefit if the authors would center their narrative more
>>concisely around the main new finding (see also next paragraph).
>>Reading the manuscript, it feels to me that the most important new result is
>>the analytical construction in terms of the "lego scars" (I am happy to be
>>convinced by the authors if they disagree with this). However, the
>>way the paper is written right now, this section almost gets lost since it is
>>far to the end of the paper, after an extensive discussion of numerical
>>findings of different types of quantum scars whose significance is less
>>clear and unexplained.

We thank the Referee for the comment, but we emphasize that our manuscript
satisfies at least two of the four criteria listed for judging the
suitability for publication SciPost Physics. Most of the findings here (the
different variety of scars, especially of type II and IIIA, their scaling
with volume especially for wider rectangles with Ly=4, and the analytic
description of the lego scars whose number diverges as one of the
dimensions of the rectangular lattice is made arbitrarily large) could not
have been anticipated from our previous work. Furthermore, the previous
work focussed primarily on ladders with Lx=2 for the QLM, and the other
varieties of scars that emerge in wider ladders of Lx=4 does not
follow in any obvious manner. The QDM, while related to the QLM, has a
completely different Hilbert space and it was again not obvious whether the
scarring mechanism discussed for the QLM in our previous
work would have survived here.

While it is true that these
results are obtained numerically,
we point out that the microscopic models at our disposal are realistic
descriptions of several many-body systems, and do not have any simple
non-interacting limit. It is almost impossible to obtain reliable information
otherwise. Moreover, while the discovery of the lego scars is extremely
interesting in the quantum dimer model which proves that such anomalous
states do exist even in the thermodynamic limit, this was only possible with
the (numerically implemented) entanglement-minimization algorithm on the
results from exact diagonalization. We feel it is important to stress the
importance of numerics in these models, in the absence of any existing
analytical methods for mid-spectrum states.

Following the Referee's suggestion, we have now emphasized these novel
aspects at multiple places in the text. We point out that, some the types
of scars in this work, have been reported for the first time ever in the
literature, and could not have been anticipated from our previous work in
any straightforward manner. A good example is type-II scars in the quantum
link model where both zero and nonzero modes of the Okin operator mix upto
a relative phase. Further, for type-III scars, the index theorem is not
directly relevant, nor is there an exponential large number of eigenstates
of Okin with non-zero integer or simple irrational eigenvalues. These
types of scars could not have been predicted from our earlier work, and we
have clearly specified this in different parts of the manuscript.

We do recognize the abundance of numerical results in Section 7 and 9.
Even though we think that such details are very useful for the community,
we do appreciate the Referee's point about getting lost in the numerical
results. We have therefore restructured Section 7. In addition to the table,
we now have sub-headings devoted to QLM and QDM, and describe the different
types of scars separately. Section 8 has added details of the analytic
understanding for the lego scars.

>> For instance, one important aspect in the study of quantum scars
>>that comes to mind is dynamics. Can the authors comment on whether it is
>>possible to find simple physical initial states that show untypical dynamics
>>due to the scars? Is there a relation between the scars that the authors
>>observe and studies of disorder-free many-body localization in such lattice
>>gauge theories? Overall, in my opinion, the authors should strongly consider
>>to restructure the paper and focus more on the analytical construction, and
>>maybe think about delegating some less important aspects to the appendix.

Thanks to the Referee for bringing this up. The cartoon states which have
a large overlap with the scar states are expected to have anomalous
dynamics for a quench Hamiltonian with λ0. This aspect
regarding anomalous dynamics from certain initial states is already
explored in the previous PRL paper in some detail, and we do not elaborate
on this further in this article.

We have also commented that our studies do not have any connection with the
disorder-free many-body localization studies of these models. The primary
point is that in those studies, the initial states are chosen such that
these form a part of different superselection sectors, each of which has its
own dynamics. In contrast, we always consider the physics within a
single superselection sector for both the QLM and the QDM.

As mentioned before, we have restructured parts of Section 7 and 8,
motivated by the comments of the Referee. We have discussed the different
scars for QDM and QLM under separate headings so that the reader can be
always aware of what types of scar occur where. We have added some
discussion in Section 5.1 as well to highlight some crucial aspects
regarding the scars and their classification. We have also added more
details about the analytic description in Section 8. However, in view of
how the results in Section 8 were obtained, we find it uncomfortable to set
up the entire discussion about the analytic results. It is only natural that
numerical tools pave an important route to analytic understanding in
strongly correlated systems, and we think our description reflects this point
of view.

>>- In the introduction, the authors mention that constrained Hilbert spaces
>>appear to play an important role in the study of quantum scars. While QLM and
>>QDM both have constrained Hilbert spaces, the constraint is even stronger in
>>the QDM. Yet, it appears that the authors find fewer scars in this case.
>>Do the authors have a physical intuition for this finding?

Thanks for raising this interesting point. At present, we do not understand
why this is the case.

>>- Can the authors explain their choice of \lambda for the presented numerical
>>results? Why is the level spacing in Fig. 6 shown for \lambda = 0.1 and 3.13,
>>but results later are shown for other values?

We wanted to numerically demonstrate that the models are non-integrable for
all the lambda values considered in the paper, and more generally for
λ O(1). However, computing the level spacing statistics for lambda
values such as 0, or 0.5, one encounters some accidental degeneracies. While
these can be easily handled, for the level spacing distribution (Fig 6), we
decided to show results for a coupling where such accidental degeneracies
do not occur.
However, as the Referee notes, the later results are presented for several
different λ values of the same order of magnitude. This is also
done to make the point that the results are robust, and do not depend on any
particular λ values since, as pointed out in the manuscript, several
of these scars stay as exact eigenstates at any λ.

>>- Regarding the level spacing in Fig. 6, the apparent deviation from GOE
>>statistics in the QDM for larger \lambda should be explained more clearly.
>>I assume for \lambda \to \infty, one expects Poisson statistics?! At the
>>same time, the authors state that the model is nonintegrable for
>>\lambda ~ 1. However, in Fig. 6 (right, bottom) it seems like increasing the
>>systems size causes the statistics to drift towards Poisson?! This can be
>>important also for the analysis of the quantum scars later in the paper.
>>If the
>>level statistics does not follow GOE behavior, the occurrence of anomalous
>>eigenstates might be less surprising.

We would like to thank the Referee for this observation, and we have now
commented on this in the main text below Fig 6. It is to be noted that
increasing the value of λ artificially causes the Hilbert space to
"fragment" into different sectors with fixed number of flippable plaquettes,
which could be responsible for the observed drift in data for the available
system sizes. However, this phenomenon is confined to only large negative
values of λ whereas all our numerical evidence for scarring is
shown for |λ|O(1) where the above mentioned effect does not
come into play.

>>- Would the analytical "lego-scar" construction also be possible for
>>multiples L_y = 4 \times n?

Yes, we believe that the "lego-scar" construction could work for systems
with Ly=4×n, as we also mention in the paper. We are pursuing
this as a line of research, and would leave it for a systematic treatment in
a future manuscript.

>>- On page 7, the authors write that they denote both models by {\cal H}_RK.
>>On page 9, they write that they refer to both models by {\cal H}.

Thanks for pointing out the inconsistent notation, this has been corrected
in this version.

>>- Are the lattices in Fig. 4 to be interpreted with PBC, i.e., are top and
>>bottom row (left and right column) equivalent?

As the Referee correctly interprets, this is indeed the case and has now
been explicitly mentioned in the figure cation of Fig 4.

>>- I appreciate the authors efforts to provide a sketch of the different types
>>of scars in Fig. 7. I must admit however, that I don't fully understand all
>>aspects, for instance does the number of arrows has a meaning?

We appreciate that the Referee points this out. We have now suitably
modified the Fig 7, and added explanatory comments both in the figure
caption and in Section 5.1 which explains Fig. 7 (and extensively
references the different panels). We hope that Section 5.1 and Fig. 7
complement each other and give a better explanation.

Please also see the List of Changes to manuscript. We hope that with these
improvements, the Referee is convinced that the manuscript meets the
standard of SciPost Physics, and can be published in its current form.
==================================================================

---

## Round 2 · Referee Report · Anonymous (Referee 2) · 2022-3-12

Strengths

1. The paper has a very nice expository discussion of quantum link models, quantum dimer models, and the relationships between them.

2. The paper has a relatively clear discussion of how to look for atypical eigenstates of the models they study numerically. There is also a comprehensive discussion of which symmetries they implement.

3. The paper presents a constructive prescription for a class of many-body scar states in the quantum dimer model.

Weaknesses

1. Apart from the "Lego scars" that have a simple construction, there is very little insight as to why the scar states discussed in this manuscript appear. Instead, they are found empirically and identified with different classes depending on how they mix zero and nonzero modes of the unperturbed Hamiltonian.

2. While a nice algorithm for extracting scar states from the spectrum is given in Sec. 6, it seems that this method only works for "Type I" scars that are composed of zero modes of the kinetic term. It was not clear to me how scar states of the other types were identified, except perhaps by brute-force calculation of, e.g., entanglement or Shannon entropies.

3. Although there is clear evidence of outliers in the entanglement and Shannon entropies for various finite-size systems, without a physical mechanism underlying their appearance it is difficult to gauge how important these states are as the system size is scaled up.

Report

I believe that the paper meets the acceptance criteria of SciPost and that it should be published with some relatively minor changes. Below I list a few questions that the authors should answer before publication of the article and discuss in the manuscript if appropriate:

1. In the discussion of Type-III scars in Sec. 5.1, the authors show a factorization of the characteristic polynomial of O_{kin}. Some aspects of these factorizations were not obvious to me--for example, the integer and simple irrational eigenvalues come in pairs with opposite signs. Are these features determined empirically, i.e. from the examples where they are found, or are they derived? If they are empirical, I think it would be good to state this explicitly. If not, it should be explained how these features are derived.

2. The entanglement entropy plots in Figs. 10, 13, 14, 21, and 22 seem to show many outlier states that are not discussed. Do the authors have any comment to make on these states? Perhaps they are related to Hilbert space fragmentation/unflippable plaquettes (see below).

3. Per the discussion in Sec. 8, it seems that the constructive prescription for the Lego scars only works in a quasi 1D limit where one of the dimensions of the square lattice is fixed to 4. Is it really not possible to extend this construction to other lattice widths/heights? If so, I think it would be good to explain why.

4. In the discussion of the Lego scars, it seems that an important ingredient is the existence of "inactive zones" that are frozen under application of the kinetic term. This seems to indicate that Hilbert space fragmentation is a necessary ingredient for these states, but I did not see any mention or discussion of this. If the authors agree with this point then I think it should be explicitly spelled out in the paper, as this leads to a simple interpretation of the lego scar states as linear combinations of a small number of product states that span a certain Krylov sector.

A list of minor notation/cosmetic changes that should be made are listed under "Requested Changes."

Requested changes

1. On page 6 the authors begin using the term "charge" without describing what it means in the context of Gauss's law. My understanding is that the charge background (and therefore the distinction between QLM and QDM) corresponds to a fixed eigenspace of the gauge operators G_r, but this is not spelled out explicitly in the discussion around Eqs. 4 and 5.

2. I was a bit confused by the notation in Eq. 6--shouldn't the sums run only over x and y, with y_0 and x_0 held fixed?

3. It should be made clearer in Fig. 4 how periodic boundary conditions are imposed, as it affects the reader's visual calculation of the winding numbers quoted in the figure. Something along the lines of, e.g., Fig. 16, where the rightmost vertical links are omitted as they coincide with the leftmost ones, would be nice.

4. Ref. 82 is a duplicate of Ref. 55.

5. Eq. 8 should cite the paper where these expressions are derived [I believe it is Phys. Rev. Lett. 110, 084101 (2013)].

6. The authors should explain why the observable defined in Eq. 11 is a good one to use for probing the typicality/atypicality of eigenstates. Does it have some physical interpretation?

7. I found Fig. 7 very confusing. It was not clear to me what the solid and dashed arrows represent. Also, the equation above the lower-right portion of the figure does not match the notation used in the text.

8. In Fig. 16, it would be useful to use different colors for the plaquettes shaded in blue that have circulation in opposite directions, as is done in Fig. 4.

  • validity: -
  • significance: -
  • originality: -
  • clarity: -
  • formatting: -
  • grammar: -

Author:  Arnab Sen  on 2022-03-29  [id 2335]

(in reply to Report 2 on 2022-03-12)

Response to Referee 2
==================================================================
We thank the Referee for taking the time and effort to carefully review our
manuscript. We appreciate that the Referee liked the expository discussion,
the identifications of scar states and the prescription
to construct many-body scar states in the dimer model. We acknowledge the
various weakness in the paper that the Referee points and have addressed them
in this version according to the suggestions offered.
Please find our responses to the specific points that were raised below.
Please also see the List of Changes to the manuscript, implemented to address
the comments.

>> The Weakness in the Paper:
>>1. Apart from the "Lego scars" that have a simple construction, there is
>>very little insight as to why the scar states discussed in this
>>manuscript appear. Instead, they are found empirically and identified with
>>different classes depending on how they mix zero and nonzero
>>modes of the unperturbed Hamiltonian.

The presence of the type-I anomalous states in these models can be
understood from the "order-by-disorder" mechanism in the Hilbert space
as we emphasize in the present manuscript. We have also tried to clarify
the role of the zero and non-zero modes in forming the other scar
varieties in Sec 5.1 now. Some of these scar varieties have been reported
for the first time (albeit numerically) in the literature to the best of our
knowledge. Of course, we do not claim to understand why the
presence of Opot gives rise to these special states apart from the analytic
construction for the lego scars. The relevant point, that other operators
could crystallize other special directions or scars, is emphasized in the
manuscript. We, however, believe that we have clearly pointed out that
local Hamiltonians with large nullspaces and addition of non-commuting terms
to such models can give rise to a rich variety of different quantum many-body
scars.

>>2. While a nice algorithm for extracting scar states from the spectrum is
>>given in Sec. 6, it seems that this method only works for "Type
>>I" scars that are composed of zero modes of the kinetic term. It was not
>>clear to me how scar states of the other types were identified,
>>except perhaps by brute-force calculation of, e.g., entanglement or
>>Shannon entropies.

Thanks for pointing this out. The algorithm can, in fact, detect all
the scars except type-II scars. We now explicitly state that in Sec. 6.
furthermore, all the scars discussed here have tell-tale energies that the
nearby eigenstates do not possess. We have also mentioned this fact in the
manuscript now.

>>3. Although there is clear evidence of outliers in the entanglement and
>>Shannon entropies for various finite-size systems, without a
>>physical mechanism underlying their appearance it is difficult to gauge
>>how important these states are as the system size is scaled up.

To address this comment, we have now added some text in the Conclusion
noting that the total number of scars obtained numerically on the finite
lattices seem to increase significantly. Given that the total number of
scars in the QLM is always greater than the QDM of the same lattice
dimensions, we find it plausible to expect that the exponentially large
bound we obtain for the QDM for Ly=4 and arbitrary Lx could also serve as
a lower bound for the total number of scars in the QLM.

>> Report:
>>1. In the discussion of Type-III scars in Sec. 5.1, the authors show a
>>factorization of the characteristic polynomial of O_{kin}. Some
>>aspects of these factorizations were not obvious to me--for example, the
>>integer and simple irrational eigenvalues come in pairs with
>>opposite signs. Are these features determined empirically, i.e. from the
>>examples where they are found, or are they derived? If they are
>>empirical, I think it would be good to state this explicitly. If not,
>>it should be explained how these features are derived.

We thank the Referee for raising this point. This feature arises due to
the fact that the eigenvalue spectrum is symmetric about E=0 for
λ=0, due to the existence of the C-operator which anticommutes
with the Hamiltonian. The irrational eigenvalues can only come as the
roots of integers which factorize in the same way as described before.
We explicitly make this statement in the main text of the paper now.

>>2. The entanglement entropy plots in Figs. 10, 13, 14, 21, and 22 seem to
>>show many outlier states that are not discussed. Do the authors
>>have any comment to make on these states? Perhaps they are related to
>>Hilbert space fragmentation/unflippable plaquettes (see below).

Thanks for the Referee for this observation. The presence of these
outlier states, to the best of our understanding, is not due to the
Hilbert space fragmentation which is in fact absent for these two models
(we address this in the next point).
It is indeed possible that there is yet another scarring mechanism beyond
what has been discussed in this manuscript for these two models.
Another possiblility for the origin of these states could be that they are
not fully "stabilized" by Opot. Perhaps a different operator, one which
does not commute with Okin can stabilize these states in
a tangible fashion. We state this possibility in the conclusion.

>>3. Per the discussion in Sec. 8, it seems that the constructive prescription
>>for the Lego scars only works in a quasi 1D limit where one of
>>the dimensions of the square lattice is fixed to 4. Is it really not
>>possible to extend this construction to other lattice widths/heights?
>>If so, I think it would be good to explain why.

Thanks to the Referee for this point, which is also raised by the other
Referee. We are optimistic that this construction can be extended to
lattices whose y-extent is a multiple of 4, but we would like
to attempt this as a separate project.

>>4. In the discussion of the Lego scars, it seems that an important
>>ingredient is the existence of "inactive zones" that are frozen under
>>application of the kinetic term. This seems to indicate that Hilbert space
>>fragmentation is a necessary ingredient for these states, but I
>>did not see any mention or discussion of this. If the authors agree with
>>this point then I think it should be explicitly spelled out in the
>>paper, as this leads to a simple interpretation of the lego scar states as
>>linear combinations of a small number of product states that
>>span a certain Krylov sector.

We thank the Referee for raising this important point and apologize that
this has not been discussed clearly in the previous version of the
manuscript. In accordance with conventional wisdom, it turns out that
neither the dimer model nor the link model in two-spatial dimension is
fragmented. We have explicitly verified this by considering all basis
states of the system on a lattice (within our computational limits) and
showing that it is possible to reach every other state in the same
symmetry sector by repeated applications of the Hamiltonian operator.
In fact, this was used to independently check the number of gauge invariant
states that was obtained from a different counting.

We have explained in detail in the text that the existence of "inactive
zones" are essentially due to the presence of the legos which contain a
superposition of two states, each having a clockwise and anticlockwise
flippable plaquette, separated by non-flippable plaquettes and the precise
sign structure. When this is the case, the action of the Hamiltonian on
such a lego creates states with alternating clockwise and anticlockwise
plaquettes that cancel each other in the first step. Without such a
cancellation, the action of the Hamiltonian causes flippable plaquettes to
appear in the "inactive zone" and the quantum caging is lost.

We have emphasized the necessity of the sign structure at multiple points
in Section 8, and hope that it is clear that such a construction does not
imply the fragmentation of Hilbert space. This is an important point,
since these models (specially the quantum dimer model) have been
extensively subject to exact diagonalization by different authors before,
and it is well known that Hilbert space fragmentation does not
happen in this model.

>> Requested Changes
>>1. On page 6 the authors begin using the term "charge" without describing
>>what it means in the context of Gauss's law. My understanding is
>>that the charge background (and therefore the distinction between QLM and
>>QDM) corresponds to a fixed eigenspace of the gauge operators
>>G_r, but this is not spelled out explicitly in the discussion around Eqs.
>>4 and 5.

The Referee is completely right, and we apologize if this was not spelt out
clearly. We have added a sentence below Eq (5) emphasizing the static
nature of the charges in the dimer model.

>>2. I was a bit confused by the notation in Eq. 6--shouldn't the sums run
only over x and y, with y_0 and x_0 held fixed?

This is correct, and the existing sentence below Eq. 6, explains it:
"The summation needs to be taken along the x-direction at y = y_0 for
vertical links to compute W_x and along
the y-direction at x = x_0 for horizontal links to compute W_y , as
shown in Fig. 4."

>>3. It should be made clearer in Fig. 4 how periodic boundary conditions are
>>imposed, as it affects the reader's visual calculation of the
>>winding numbers quoted in the figure. Something along the lines of, e.g.,
>>Fig. 16, where the rightmost vertical links are omitted as they
>>coincide with the leftmost ones, would be nice.

We have added an explicit statement in the figure caption emphasizing how
the periodic boundary conditions are imposed on both directions.

>>4. Ref. 82 is a duplicate of Ref. 55.

This has been corrected.

>>5. Eq. 8 should cite the paper where these expressions are derived
>>[I believe it is Phys. Rev. Lett. 110, 084101 (2013)].

We include the citation as requested.

>>6. The authors should explain why the observable defined in Eq. 11 is a
>>good one to use for probing the typicality/atypicality of
>>eigenstates. Does it have some physical interpretation?

The observable in Eq. 11 can be interpreted as the sum of local
correlation functions of electric flux operators. This gauge invariant
observable is an operator that can be the most easily measured, both
theoretically, and if ever implemented, experimentally, on potential
cold-atom setups.

>>7. I found Fig. 7 very confusing. It was not clear to me what the solid and
>>dashed arrows represent. Also, the equation above the lower
->>right portion of the figure does not match the notation used in the text.

We appreciate that the Referee points this out. We have now suitably
modified the Fig 7, and added explanatory comments both in the figure
caption and in Section 5.1 which explains Fig. 7
(and extensively references the different panels). We hope that
Section 5.1 and Fig. 7 complement
each other and give a better explanation.

>>8. In Fig. 16, it would be useful to use different colors for the plaquettes
>>shaded in blue that have circulation in opposite directions,
>>as is done in Fig. 4.

Thanks for pointing this out. Regretfully, this change is not so trivial
to implement at the moment. These
figures are directly printed out from an optimized python code. We would
like to request the Referee to kindly
accept the figure as it is.

---

## Round 3 · Referee Report · Anonymous (Referee 2) · 2022-4-7

Report

The authors have sufficiently addressed my comments and clarified some points of confusion from my first reading of the paper.

---

## Round 3 · Referee Report · Anonymous (Referee 1) · 2022-4-7

Report

I thank the authors for their reply and additional clarifications, as well as
their efforts to improve the presentation of their manuscript. While I still
think that the paper would benefit from a more concise presentation, I respect
the authors' choice to keep their format. As already written in my previous
report, I believe that the paper contains new results that can be of interest to
researchers working in the field. Taking into account the author's replies to
both referee reports, I recommend publication of the manuscript in SciPost
Physics as well as SciPost Physics Core, at the Editor's discretion.

---

## Round 3 · Author Response

Dear Editor-in-charge,

We would like to thank you very much for facilitating the peer-review of this work. We are happy to note that both the referees, as well as yourself, seem to be positive about the publication of the manuscript in SciPost Physics with minor revisions. In the current version, we have addressed all the concerns of the referees and provided adequate justification for publication of this manuscript in SciPost Physics, over SciPost Core. A list of changes to the manuscript is also attached. We hope that you will consider our reasoning favourably and find the modified manuscript suitable for publication in SciPost Physics.

We look forward to hearing from you. Best regards Saptarshi Biswas, Debasish Banerjee, Arnab Sen

Justification for publication in SciPost Physics:

Among the listed criteria for publication in the SciPost physics, we consider the third and fourth to be already satisfied by our manuscript. We elaborate this below.

Criteria 3: "Open a new pathway in an existing or a new research direction, with clear potential for multipronged follow-up work"

--- Our results not only establish the validity of the "order-by-disorder" phenomenon in Hilbert space in a class of U(1) quantum link and dimer models, but also identify different types of scars, whose number seem to increase with lattice sizes. This deserves further work in the future using more advanced numerical methods to approach the two-dimensional limit. Further, the research involving the construction of exact lego scars can also be extended to larger lattices. There are a class of several interesting lattice gauge theories in higher dimensions with exponentially large nullspaces --- the research presented here should also be extended to those models to look for quantum many-body scars there. It is useful to highlight that this scarring mechanism is fundamentally distinct from "PXP type scars" and arise when additional non-commuting interactions are added to local Hamiltonians with large nullspaces protected by an index theorem. This provides a new research direction in the area of quantum many-body scars.

Criterial 4: "Provide a novel and synergetic link between different research areas."

--- Current research in condensed matter and lattice gauge theories are both positively impacted by techniques and methods of quantum information, boosting efforts to build experimental realizations for physically relevant models in the two fields. Our research uses models, considered paradigmatic in both communities, and suggests novel phenomena that can be studied and verified with the help of quantum simulators.

Response to Referee 1

We thank the Referee for taking the time and effort to carefully review our manuscript. We are happy that they find the introduction comprehensive, the numerics advanced, and the analytic understanding of some scars noteworthy. We also appreciate pointing out the weak points of the current paper, and we have improved and streamlined the presentation considerably to meet the standars of SciPost physics. Please find our responses to the specific points that were raised. Please also see the List of Changes to the manuscript, implemented to address the comments.

I support to proceed towards the publication of the manuscript. While the manuscript clearly fulfills all acceptance criteria of Scipost Physics Core, I am unsure however if the novelty of the results warrants publication in Scipost Physics, not least as many of the findings presented here appear to have already been anticipated in a previous work of the authors (as also stated in the abstract). Moreover, the manuscript presents an abundance of numerical results, but often it is hard to follow the different parameter choices (I appreciate the authors' attempt to support understandability by providing overview tables) and, more importantly, physical interpretation of the obtained results is scarce. In my opinion, it should be emphasized more clearly what are the new results that substantially go beyond their previous work and their importance should be explained. Likewise, the manuscript and its readers would benefit if the authors would center their narrative more concisely around the main new finding (see also next paragraph). Reading the manuscript, it feels to me that the most important new result is the analytical construction in terms of the "lego scars" (I am happy to be convinced by the authors if they disagree with this). However, the way the paper is written right now, this section almost gets lost since it is far to the end of the paper, after an extensive discussion of numerical findings of different types of quantum scars whose significance is less clear and unexplained.

We thank the Referee for the comment, but we emphasize that our manuscript satisfies at least two of the four criteria listed for judging the suitablitly for publication SciPost Physics. Most of the findings here (the different variety of scars, especially of type II and IIIA, their scaling with volume especially for wider rectangles with Ly=4, and the analytic description of the lego scars whose number diverges as one of the dimensions of the rectangular lattice is made arbitrarily large) could not have been anticipated from our previous work. Furthermore, the previous work focussed primarily on ladders with Lx=2 for the QLM, and the other varieties of scars that emerge in wider ladders of Lx=4 does not follow in any obvious manner. The QDM, while related to the QLM, has a completely different Hilbert space and it was again not obvious whether the scarring mechanism discussed for the QLM in our previous work would have survived here.

While it is true that these results are obtained numerically, we point out that the microscopic models at our disposal are realistic descriptions of several many-body systems, and do not have any simple non-interacting limit. It is almost impossible to obtain reliable information otherwise. Moreover, while the discovery of the lego scars is extremely interesting in the quantum dimer model which proves that such anomalous states do exist even in the thermodynamic limit, this was only possible with the (numerically implemented) entanglement-minimization algorithm on the results from exact diagonalization. We feel it is important to stress the importance of numerics in these models, in the absence of any existing analytical methods for mid-spectrum states.

Following the Referee's suggestion, we have now emphasized these novel aspects at multiple places in the text. We point out that, some the types of scars in this work, have been reported for the first time ever in the literature, and could not have been anticipated from our previous work in any straightforward manner. A good example is type-II scars in the quantum link model where both zero and nonzero modes of the Okin operator mix upto a relative phase. Further, for type-III scars, the index theorem is not directly relevant, nor is there an exponential large number of eigenstates of Okinm with non-zero integer or simple irrational eigenvalues. These types of scars could not have been predicted from our earlier work, and we have clearly specified this in different parts of the manuscript.

We do recognize the abundance of numerical results in Section 7 and 9. Even though we think that such details are very useful for the community, we do appreciate the Referee's point about getting lost in the numerical results. We have therefore restructured Section 7. In addition to the table, we now have sub-headings devoted to QLM and QDM, and describe the different types of scars separately. Section 8 has added details of the analytic understanding for the lego scars.

For instance, one important aspect in the study of quantum scars that comes to mind is dynamics. Can the authors comment on whether it is possible to find simple physical initial states that show untypical dynamics due to the scars? Is there a relation between the scars that the authors observe and studies of disorder-free many-body localization in such lattice gauge theories? Overall, in my opinion, the authors should strongly consider to restructure the paper and focus more on the analytical construction, and maybe think about delegating some less important aspects to the appendix.

Thanks to the Referee for bringing this up. The cartoon states which have a large overlap with the scar states are expected to have anomalous dynamics for a quench Hamiltonian with $\lambda \neq 0$. This aspect regarding anomalous dynamics from certain initial states is already explored in the previous PRL paper in some detail, and we do not elaborate on this further in this article.

We have also commented that our studies do not have any connection with the disorder-free many-body localization studies of these models. The primary point is that in those studies, the initial states are chosen such that these form a part of different superselection sectors, each of which has its own dynamics. In contrast, we always consider the physics within a single superselection sector for both the QLM and the QDM.

As mentioned before, we have restructured parts of Section 7 and 8, motivated by the comments of the Referee. We have discussed the different scars for QDM and QLM under separate headings so that the reader can be always aware of what types of scar occur where. We have added some discussion in Section 5.1 as well to highlight some crucial aspects regarding the scars and their classification. We have also added more details about the analytic description in Section 8. However, in view of how the results in Section 8 were obtained, we find it uncomfortable to set up the entire discussion about the analytic results. It is only natural that numerical tools pave an important route to analytic understanding in strongly correlated systems, and we think our description refects this point of view.

  • In the introduction, the authors mention that constrained Hilbert spaces appear to play an important role in the study of quantum scars. While QLM and QDM both have constrained Hilbert spaces, the constraint is even stronger in the QDM. Yet, it appears that the authors find fewer scars in this case. Do the authors have a physical intuition for this finding?

Thanks for raising this interesting point. At present, we do not understand why this is the case.

  • Can the authors explain their choice of \lambda for the presented numerical results? Why is the level spacing in Fig. 6 shown for \lambda = 0.1 and 3.13, but results later are shown for other values?

We wanted to numerically demonstrate that the models are non-intergrable for all the lambda values considered in the paper, and more generally for $\lambda ~ O(1)$. However, computing the level spacing statistics for lambda values such as 0, or 0.5, one encounters some accidental degeneracies. While these can be easily handled, for the level spacing distribution (Fig 6), we decided to show results for a coupling where such accidental degeneracies do not occur. However, as the Referee notes, the later results are presented for several different $\lambda$ values of the same order of magnitude. This is also done to make the point that the results are robust, and do not depend on any particular $\lambda$ values since, as pointed out in the manuscript, several of these scars stay as exact eigenstates at any $\lambda$.

  • Regarding the level spacing in Fig. 6, the apparent deviation from GOE statistics in the QDM for larger \lambda should be explained more clearly. I assume for \lambda \to \infty, one expects Poisson statistics?! At the same time, the authors state that the model is nonintegrable for \lambda ~ 1. However, in Fig. 6 (right, bottom) it seems like increasing the systems size causes the statistics to drift towards Poisson?! This can be important also for the analysis of the quantum scars later in the paper. If the level statistics does not follow GOE behavior, the occurrence of anomalous eigenstates might be less surprising.

We would like to thank the Referee for this observation, and we have now commented on this in the main text below Fig 6. It is to be noted that increasing the value of $\lambda$ artificially causes the Hilbert space to "fragment" into different sectors with fixed number of flippable plaquettes, which could be responsible for the observed drift in data for the available system sizes. However, this phenomenon is confined to only large negative values of $\lambda$ whereas all our numerical evidence for scarring is shown for $|\lambda| \sim O(1)$ where the above mentioned effect does not come into play.

  • Would the analytical "lego-scar" construction also be possible for multiples L_y = 4 \times n?

Yes, we believe that the "lego-scar" construction could work for systems with $L_y = 4 \times n$, as we also mention in the paper. We are pursuing this as a line of research, and would leave it for a systematic treatment in a future manuscript.

  • On page 7, the authors write that they denote both models by {\cal H}_RK. On page 9, they write that they refer to both models by {\cal H}.

Thanks for pointing out the inconsistent notation, this has been corrected in this version.

  • Are the lattices in Fig. 4 to be interpreted with PBC, i.e., are top and bottom row (left and right column) equivalent?

As the Referee correctly interprets, this is indeed the case and has now been explicitly mentioned in the figure cation of Fig 4.

  • I appreciate the authors efforts to provide a sketch of the different types of scars in Fig. 7. I must admit however, that I don't fully understand all aspects, for instance does the number of arrows has a meaning?

We appreciate that the Referee points this out. We have now suitably modified the Fig 7, and added explanatory comments both in the figure caption and in Section 5.1 which explains Fig. 7 (and extensively references the different panels). We hope that Section 5.1 and Fig. 7 complement each other and give a better explanation.

Please also see the List of Changes to manuscript. We hope that with these improvements, the Referee is convinced that the manuscript meets the standard of SciPost Physics, and can be published in its current form.

Response to Referee 2

We thank the Referee for taking the time and effort to carefully review our manuscript. We appreciate that the Referee liked the expository discussion, the identifications of scar states and the prescription to construct many-body scar states in the dimer model. We acknowledge the various weakness in the paper that the Referee points and have addressed them in this version according to the suggestions offered.

Please find our responses to the specific points that were raised below. Please also see the List of Changes to the manuscript, implemented to address the comments.

The Weakness in the Paper:

1. Apart from the "Lego scars" that have a simple construction, there is very little insight as to why the scar states discussed in this manuscript appear. Instead, they are found empirically and identified with different classes depending on how they mix zero and nonzero modes of the unperturbed Hamiltonian.

The presence of the type-I anomalous states in these models can be understood from the "order-by-disorder" mechanism in the Hilbert space as we emphasize in the present manuscript. We have also tried to clarify the role of the zero and non-zero modes in forming the other scar varieties in Sec 5.1 now. Some of these scar varieties have been reported for the first time (albeit numerically) in the literature to the best of our knowledge. Of course, we do not claim to understand why the presence of Opot gives rise to these special states apart from the analytic construction for the lego scars. The relevant point, that other operators could crystallize other special directions or scars, is emphasized in the manuscript. We, however, believe that we have clearly pointed out that local Hamiltonians with large nullspaces and addition of non-commuting terms to such models can give rise to a rich variety of different quantum many-body scars.

2. While a nice algorithm for extracting scar states from the spectrum is given in Sec. 6, it seems that this method only works for "Type I" scars that are composed of zero modes of the kinetic term. It was not clear to me how scar states of the other types were identified, except perhaps by brute-force calculation of, e.g., entanglement or Shannon entropies.

Thanks for pointing this out. The algorithm can, in fact, detect all the scars except type-II scars. We now explicitly state that in Sec. 6. Furthermore, all the scars discussed here have tell-tale energies that the nearby eigenstates do not possess. We have also mentioned this fact in the manuscript now.

3. Although there is clear evidence of outliers in the entanglement and Shannon entropies for various finite-size systems, without a physical mechanism underlying their appearance it is difficult to gauge how important these states are as the system size is scaled up.

To address this comment, we have now added some text in the Conclusion noting that the total number of scars obtained numerically on the finite lattices seem to increase significantly. Given that the total number of scars in the QLM is always greater than the QDM of the same lattice dimensions, we find it plausible to expect that the exponentially large bound we obtain for the QDM for Ly=4 and arbitrary Lx could also serve as a lower bound for the total number of scars in the QLM.

Report: 1. In the discussion of Type-III scars in Sec. 5.1, the authors show a factorization of the characteristic polynomial of O_{kin}. Some aspects of these factorizations were not obvious to me--for example, the integer and simple irrational eigenvalues come in pairs with opposite signs. Are these features determined empirically, i.e. from the examples where they are found, or are they derived? If they are empirical, I think it would be good to state this explicitly. If not, it should be explained how these features are derived.

We thank the Referee for raising this point. This feature arises due to the fact that the eigenvalue spectrum is symmetric about E=0 for $\lambda=0$, due to the existance of the C-operator which anticommutes with the Hamiltonian. The irrational eigenvalues can only come as the roots of integers which factorize in the same way as described before. We explicitly make this statement in the main text of the paper now.

2. The entanglement entropy plots in Figs. 10, 13, 14, 21, and 22 seem to show many outlier states that are not discussed. Do the authors have any comment to make on these states? Perhaps they are related to Hilbert space fragmentation/unflippable plaquettes (see below).

Thanks for the Referee for this observation. The presence of these outlier states, to the best of our understanding, is not due to the Hilbert space fragmentation which is in fact absent for these two models (we address this in the next point). It is indeed possible that there is yet another scarring mechanism beyond what has been discussed in this manuscript for these two models. Another possiblility for the origin of these states could be that they are not fully "stabilized" by Opot. Perhaps a different operator, one which does not commute with Okin can stablize these states in a tangible fashion. We state this possibility in the conclusion.

3. Per the discussion in Sec. 8, it seems that the constructive prescription for the Lego scars only works in a quasi 1D limit where one of the dimensions of the square lattice is fixed to 4. Is it really not possible to extend this construction to other lattice widths/heights? If so, I think it would be good to explain why.

Thanks to the Referee for this point, which is also raised by the other Referee. We are optimistic that this construction can be extended to lattices whose y-extent is a multiple of 4, but we would like to attempt this as a separate project.

4. In the discussion of the Lego scars, it seems that an important ingredient is the existence of "inactive zones" that are frozen under application of the kinetic term. This seems to indicate that Hilbert space fragmentation is a necessary ingredient for these states, but I did not see any mention or discussion of this. If the authors agree with this point then I think it should be explicitly spelled out in the paper, as this leads to a simple interpretation of the lego scar states as linear combinations of a small number of product states that span a certain Krylov sector.

We thank the Referee for raising this important point and apologize that this has not been discussed clearly in the previous version of the manuscript. In accordance with conventional wisdom, it turns out that neither the dimer model nor the link model in two-spatial dimension is fragmented. We have explicitly verified this by considering all basis states of the system on a lattice (within our computational limits) and showing that it is possible to reach every other state in the same symmetry sector by repeated applications of the Hamiltonian operator. In fact, this was used to independently check the number of gauge invariant states that was obtained from a different counting.

We have explained in detail in the text that the existence of "inactive zones" are essentially due to the presence of the legos which contain a superposition of two states, each having a clockwise and anticlockwise flippable plaquette, separated by non-flippable plaquettes and the precise sign structure. When this is the case, the action of the Hamiltonian on such a lego creates states with alternating clockwise and anticlockwise plaquettes that cancel each other in the first step. Without such a cancellation, the action of the Hamiltonian causes flippable plaquettes to appear in the "inactive zone" and the quantum caging is lost.

We have emphasized the necessity of the sign structure at multiple points in Section 8, and hope that it is clear that such a construction does not imply the fragmentation of Hilbert space. This is an important point, since these models (specially the quantum dimer model) have been extensively subject to exact diagonalization by different authors before, and it is well known that Hilbert space fragmentation does not happen in this model.

Requested Changes

1. On page 6 the authors begin using the term "charge" without describing what it means in the context of Gauss's law. My understanding is that the charge background (and therefore the distinction between QLM and QDM) corresponds to a fixed eigenspace of the gauge operators G_r, but this is not spelled out explicitly in the discussion around Eqs. 4 and 5.

The Referee is completely right, and we apologize if this was not spelt out clearly. We have added a sentence below Eq (5) emphasizing the static nature of the charges in the dimer model.

2. I was a bit confused by the notation in Eq. 6--shouldn't the sums run only over x and y, with y_0 and x_0 held fixed?

This is correct, and the existing sentence below Eq. 6, explains it: "The summation needs to be taken along the x-direction at y = y_0 for vertical links to compute W_x and along the y-direction at x = x_0 for horizontal links to compute W_y , as shown in Fig. 4."

3. It should be made clearer in Fig. 4 how periodic boundary conditions are imposed, as it affects the reader's visual calculation of the winding numbers quoted in the figure. Something along the lines of, e.g., Fig. 16, where the rightmost vertical links are omitted as they coincide with the leftmost ones, would be nice.

We have added an explicit statement in the figure caption emphasizing how the periodic boundary conditions are imposed on both directions.

4. Ref. 82 is a duplicate of Ref. 55.

This has been corrected.

5. Eq. 8 should cite the paper where these expressions are derived [I believe it is Phys. Rev. Lett. 110, 084101 (2013)].

We include the citation as requested.

6. The authors should explain why the observable defined in Eq. 11 is a good one to use for probing the typicality/atypicality of eigenstates. Does it have some physical interpretation?

The observable in Eq. 11 can be interpreted as the sum of local correlation functions of electric flux operators. This gauge invariant observable is an operator that can be the most easily measured, both theoretically, and if ever implemented, experimentally, on potential cold-atom setups.

7. I found Fig. 7 very confusing. It was not clear to me what the solid and dashed arrows represent. Also, the equation above the lower-right portion of the figure does not match the notation used in the text.

We appreciate that the Referee points this out. We have now suitably modified the Fig 7, and added explanatory comments both in the figure caption and in Section 5.1 which explains Fig. 7 (and extensively references the different panels). We hope that Section 5.1 and Fig. 7 complement each other and give a better explanation.

8. In Fig. 16, it would be useful to use different colors for the plaquettes shaded in blue that have circulation in opposite directions, as is done in Fig. 4.

Thanks for pointing this out. Regretfully, this change is not so trivial to implement at the moment. These figures are directly printed out from an optimized python code. We would like to request the Referee to kindly accept the figure as it is.

---

## Round 3 · List of Changes

---A line has been added in the abstract to highlight the tell-tale energies
of these quantum scars.
---Expanded the introduction a bit (Sec 1) to highlight certain aspects a bit
more.
---Added a line in Sec 2.
---Added a line in caption of Fig. 4.
---Added a couple of lines towards the end of Sec. 4 to explain a technical
point.
---Added a line after Eq. 11.
---Section 5.1 expanded a bit more to explain the different scar varities in a
better manner.
---The schematic figure (Fig. 7) changed a bit and its caption expanded.
---Added a couple of lines towards the end of Sec. 6 to explain a technical
point.
---Added a line in Sec. 7 and divided the text in two separate subsections.
---Expanded Section 8 a bit more to explain how the inactive regions are
tied to the sign structure inside the active regions.
---Expanded the "Conclusions and Outlook" section to highlight a few points in
a better manner.
---Some new references have been added.

---

## Editorial Decision

published